# The NIEER AVHRR snow cover extent product over China – A long-term daily snow record for regional climate research

Xiaohua Hao[1,2], Guanghui Huang[3], Tao Che[1,2], Wenzheng Ji[1], Xingliang Sun[1,4],Qin Zhao[1],Hongyu Zhao[1], Jian Wang[1,2], Hongyi Li[1.2], Qian Yang[5].

[1]Heihe Remote Sensing Experimental Research Station, Northwest Institute of Eco-Environment and Resources, Chinese Academy of Sciences, Lanzhou 730000, China
[2] Key Laboratory of Remote Sensing of Gansu Province, Northwest Institute of Eco-Environment and Resources, Chinese Academy of Sciences, Lanzhou 730000, China
[3]College of Earth and Environmental Sciences, Lanzhou University, Lanzhou 730000, China
[4]Engineering Laboratory for National Geographic State Monitoring, Lanzhou Jiaotong University, Lanzhou 730070, China
[5]School of Geomatics and Prospecting Engineering, Jilin Jianzhu University, Changchun 130118, China

*Correspondence to:* Xiaohua Hao (haoxh@lzb.ac.cn)

**Abstract.** Using the Google Earth Engine (GEE) platform, a long-term AVHRR snow cover extent (SCE) product from 1981 until 2019 over China has been generated by the snow research team in the Northwest Institute of Eco-Environment and Resources (NIEER), Chinese Academy of Sciences. The new NIEER product has the spatial resolution of 5-km and the daily temporal resolution, and is a completely gap-free product, which is produced through a series of processes such as the quality control, cloud detection, snow discrimination and gap-filling. A comprehensive validation with reference to ground snow-depth measurements during snow seasons in China revealed the overall accuracy is 87.4%, the producer's accuracy was 81.0% the user's accuracy was 81.3%, and the Cohen's kappa value was 0.717. Another validation with reference to higher-resolution snow maps derived from Landsat-5 Thematic Mapper (TM) images demonstrates an overall accuracy of 89.4%, a producer's accuracy of 90.2%, a user's accuracy of 96.1%, and a Cohen's kappa value of 0.713. These accuracies were significantly higher than those of currently existing AVHRR products. For example, compared with the well-known JASMES AVHRR product, the overall accuracy increased approximately 15 percent, the omission error dropped from nearly 40% to 19.7%, the commission error dropped from 31.9% to 21.3%, and the CK value increased by more than 114%. The new AVHRR product is now already available at https://dx.doi.org/10.11888/Snow.tpdc.271381 (Hao et al. 2021).



## 1 Introduction

Snow cover is closely bound up with our climate. On one hand, owing to snow's unique optical properties (high albedo) it can affect the surface radiation budget severely, and thereby our climate systems

significantly (Warren, 1982; Huang et al. 2019). On the other hand, changes in climate in turn affect global and regional snow covers. With the continuous warming of the global climate, snow cover on the Earth has been shrinking evidently over these decades (Barnett et al., 2005; Bormann et al., 2018). Long-term snow cover data, therefore, is not only of particular importance for climate research but also an indispensable indicator of climate change.

Remote sensing is a widely used tool for monitoring snow cover extent (SCE) globally and regionally at various spatial and temporal resolutions (Konig et al., 2001; Dozier and Painter, 2004; Frei et al., 2012; Wang et al., 2014) since the beginning of the satellite era in the 1960s. The Northern Hemisphere Weekly Snow Cover and Sea Ice Extent (NHSCE) product provide weekly SCE with spatial resolutions of about 190 km from 1966 to 1997 (Robinson et al., 1993). Although the time coverage is long, the NHSCE

product has a low spatio-temporal resolution, hand-drawn snow line maps, and incomplete spatial coverage due to swath gaps or cloud obscuration, largely restricting its application in climate research. With the development of satellite sensors, SCE products with high spatial resolution throughout China have been issued in the last decades, such as the Interactive Multi-sensor Snow and Ice Mapping System (IMS), which provides daily SCE with spatial resolutions of 24 km, 4 km, and 1km from 1997 to the

present (Helfrich et al., 2007; Ramsay, 1998). The Moderate Resolution Imaging Spectroradiometer (MODIS) provides daily SCE with a spatial resolution of 500 m from 2000 to the present (Hall et al., 2002; Riggs et al., 2017). The polar orbit meteorological satellite of the Fengyun daily SCE products has a spatial resolution of 1 km from 2003 to the present (Min et al., 2021). These SCE datasets have good quality with a high spatio-temporal resolution, but their short period is insufficient to create a

climatological baseline of snow cover.

The Japan Aerospace Exploration Agency (JAXA) recently issued the long-term snow cover extent (SCE) product JASMES with a spatial resolution of 5 km throughout the Northern Hemisphere. This product consists of satellite-derived daily, weekly, and half-monthly averaged global snow covers derived from 5 km resampled radiance data of AVHRR Global Area Coverage (GAC) radiance data onboard NOAA

series satellites (1978-2001) and MODIS onboard Terra & Aqua satellites (2000–the present) (Hori et

al., 2017). Although the JASMES product presented a long time series and significantly enhanced spatial and temporal resolution, several shortcomings have been found. (1) The JASMES product uses AVHHR before 2000 and MODIS data after 2000. Although calibrated by the authors, the bandwidths of the two sensors are not consistent, and using the same algorithms for both can cause discontinuities in the data.

(2) Previous work showed that the JASMES snow product has an excessive cloud mask, which would cause a considerable number of snow pixels to be misidentification as cloud pixels (Wang et al., 2018). (3) JASMES snow algorithm tended to underestimate snow in China, especially on the Qinghai-Tibet Plateau (Wang et al., 2018). (4)Finally, JASMES SCE exhibits incomplete spatial coverage caused by clouds and data gaps. These shortcomings limit its application in snow monitoring and climate studies in

China. Thus, China still lacks a high-quality, long-term SCE product with complete spatial coverage for climate research.

Therefore, a new daily 5-km gap-free AVHRR snow cover extent product for China was produced based on the Google Earth Engine platform from 1981 to 2019. The new product provides a long time series of SCE with high quality for China and makes six improvements. (1) The Climate Data Record (CDR) of

AVHRR Surface Reflectance (SR) is used as a data source after 2000 rather than MODIS to ensure product continuity. (2) Considering sensor attenuation of Band 11 before and after 2000, the algorithm chooses different training samples and discriminant thresholds separately. (3) An improved cloud detection test and new thresholds are obtained by a volume of training data,which can solve the snow/cloud confusion. (4) A multi-level decision tree for snow discrimination algorithm is applied,

which significantly improved snow discrimination accuracy. (5) Improved gap-filling strategies are adopted to obtain complete snow coverage. (6) Land surface temperature reanalysis is used to exclude the false snow identification. Due to these improvements, the new AVHRR SCE product may serve as a baseline record for climate and other related applications.

## 2 Datasets and processing

**2.1 AVHRR surface reflectance CDR**

The NOAA Climate Data Record (CDR) of AVHRR Surface Reflectance Version 4 (AVHRR SR V4) was used as basic input data. AVHRR SR V4 is generated using AVHRR Global Area Coverage (GAC) Level 1b data through geolocation, calibration, and atmospheric correction, and has latitudinal and





longitudinal dimensions of 3600×7200, covering the globe at 0.05 ° spatial resolution (Vermote et al.,
2014). The dataset contains surface reflectance, brightness, temperatures, and quality control flags for
the period between June 24, 1981, and May 16, 2019. All AVHRR SR V4 images were processed with
the GEE platform. The reflectance, brightness, and temperature data were described in Table 1. The
quality control flags are summarized in Table 3.

**2.2 Landsat-5 TM snow map**

This study used two groups of Landsat-5 Thematic Mapper (TM) maps across China from 1985-2013.
The first group was used as "true" values to acquire the training data of AVHRR surface reflectance. TM
snow maps were produced by the improved "SNOMAP" algorithm developed by Chen et al. (2020) for
the snow season (beginning on November 1 through March 31 of the following year). Each map
contained three classes, namely snow, non-snow, and cloud. Considering sensor attenuation before and
after 2000, the algorithm chose different TM images separately. Table 2 shows the number of Landsat-5
TM scenes used for training before and after 2000. The second group of maps was used as true values to
evaluate the AVHRR SCE product. A total of 8 Landsat-5 TM snow maps were used as the validation
dataset (Fig.1). The training and validating samples were evenly distributed across China's main
seasonally snow-covered areas to ensure reliability and representativeness.

**2.3 AVHRR Training Samples**

Snow and non-snow training samples from the AVHRR were generated from spatially and temporally
(same day) collocated AVHRR surface reflectance along with the Landsat-5 snow maps. Cloud training
samples came from AVHRR surface reflectance with Landsat-5 cloud flags during summer (June 1 to
August 31). The training samples before 2000 included 71.7 thousand snow samples, 80.4 thousand non-
snow samples, and 8.3 thousand cloud samples. Samples after 2000 included 7.3 million snow samples,
8.4 million non-snow samples, and 4.4 thousand cloud samples.

**2.4 Ground snow-depth measurements**

Ground snow-depth measurements provided by the China Meteorological Administration (CMA) were
used to validate the AVHRR SCE products. Daily snow depth was measured near the stations using a
professional meter ruler. All measurements were conducted at 08:00 Beijing time when the fractional



snow cover in the field of view was more than 50% (C.M.A, 2003). Validation CMA stations were carefully selected because too many non-snow samples can affect the accuracy of assessment. To ensure the validation reliability, the selected CMA stations had ≥ 20 days with true snow (>1cm) at the CMA site per snow season (Metsämäki, 2016). Finally, a total of 191 meteorological stations at 38-year periods

(from 1981 to 2019; Fig.1) were used to validate the AVHRR SCE products. The available CMA stations were evenly distributed across the three major seasonally snow-covered regions in China, including North Xinjiang, Northeast China, and the Qinghai-Tibet Plateau.

### 2.5 Ancillary data

The data set of long-term daily snow depth in China is available at http://data.tpdc.ac.cn. It provides daily,

0.25-degree snow-depth data for China from 1979 to 2020. It was generated by Che et al. (2008) and Dai et al. (2015) using an inter-sensor calibration of multiple satellites' passive-microwave observations. This data set was used as a supplement to the gap-filling strategies. We used the land surface temperature (LST) daily product to alleviate the cloud/snow confusion generated by averaging the hourly ERA 5 land climate reanalysis dataset on the GEE platform (Muñoz Sabater, 2019). Digital Elevation Model (DEM)

data were used as auxiliary data in the cloud and snow discrimination algorithm, mask, and validation. The SRTM DEM product has an original resolution of 90 m and is also available on the GEE. To match with AVHRR products, these products were resampled or aggregated into 5 km.

### 3    Methodology

Figure 2 shows the different steps in the generation of the NIEER AVHRR SCE product. Starting with

AVHRR surface reflectance version 4 (AVHRR SR V4) data on the GEE platform, valid observations were selected first by the quality control flags of AVHRR SR V4. Then, an improved cloud detection algorithm was developed to distinguish between cloudy pixels, water pixels, and clear pixels. Third, clear pixels were determined as snow-covered or not by a multi-level decision tree, generating a set of AVHRR preliminary SCE records. Forth, the gaps caused by clouds or invalid observations in the preliminary

SCE record were filled with a set of gap-filling techniques, including HRMF-based interpolation and snow-depth interpolation. Finally, postprocessing based on land surface temperature and DEM was conducted to exclude false snow identifications.



### 3.1 Quality control of AVHRR

Only observations valid in all AVHRR channels were employed to directly generate SCE records by using the quality control bit flags of AVHRR SR V4 (Table 3). In the process, the invalid pixels were regarded as gap pixels.

### 3.2 Cloud detection algorithm

In this study, we could not directly adopt the cloudy flags of AVHRR SR V4 due to the obvious cloud

overestimation (Chen et al., 2018).

As stated by previous studies (Hori et al., 2007; Hori et al., 2017; Stamnes et al., 2007; Yamanouchi et al., 1987), the following eight variables were used in the cloud detection test: SR1, SR2, SR3, BT11, the reflectance differences between SR1 and SR2 (SR1-SR2), the brightness temperature (BT) differences between BT37 and BT11 (BT37-BT11), the BT differences between BT11 and BT12 (BT11-BT12), and

the normalized difference vegetation index (NDVI). The calculation of the NDVI is based on formula (1). For cloud detection, "BT37-BT11" was used as the primary test.

$$NDVI = \frac{SR2 - SR1}{SR1 + SR2} \quad , \tag{1}$$

We adopted the cloud test scheme by Hori et al. (2017), but the critical threshold value of BT37-BT11 was adjusted. As earlier thresholds of BT37-BT11 used a stronger cloud discrimination algorithm and

ignored the cloud/snow confusion problem, further optimization was needed to minimize misclassification and the omission of clouds. Therefore, we focused on optimizing the cloud algorithm thresholds. Using the Landsat-5 TM maps for the true values, we obtained the frequency distribution characteristics of BT37-BT11 for cloud and snow samples from AVHRR SR. Table 4 shows the cloud discrimination schemes, with ten cloud detection schemes and four non-cloud schemes. With A1 type as

an example, Fig. 3 shows the optimal BT37-BT11 determination scheme. Fig. 3 (a) presents the BT37-BT11 frequency distribution of cloud and snow training samples from AVHRR before 2000, and Fig. 3 (b) presents the variation of the overall accuracy at different BT37-BT11 thresholds. Optimum accuracy (84.76%) occurred at the cross-point of snow and cloud frequency distributions, with a BT37-BT11 threshold of 14.5 K. This cross-point also represents a compromise for cloud omission (10.49%) and

commission error (19.92%). Thus, the final threshold value was 14.5 K according to the optimal OA, which means that a pixel is classified as a cloud when BT37-BT11>14.5K. Following the same procedure,



the optimal BT37-BT11 thresholds were obtained from AVHRR data before and after 2000, as listed in Table 4.

### 3.3 Snow discrimination algorithm

According to previous snow classifications with AVHHR data (Hori et al., 2007; Hori et al., 2017; Stamnes et al., 2007; Yamanouchi et al., 1987), snow discrimination test variables included SR1, BT11, the reflectance ratio between SR3 and SR2 (SR3/SR2), reflectance differences between SR3 and SR2 (SR3-SR2), NDVI, the normalized difference snow index (NDSI), and BT differences between BT11 and BT12 (BT11-BT12). For snow discrimination, the NDSI was one of the primary tests. The NDSI is

usually calculated using the Green (around a wavelength of 0.50μm) and shortwave infrared (around a wavelength of 1.60 μm) bands. As there were no shortwave infrared observations around 1.60 μm in AVHRR SR V4, we used the reflectance at 3.7 μm for an NDSI-like calculation, following Hori et al. (2017). The calculation of NDSI is shown in formula (2).

$$NDSI = \frac{SR1 - SR3}{SR1 + SR3} \quad , \tag{2}$$

To improve the snow discrimination under clear-skies, all decision rules were re-adjusted according to the training samples from high-resolution snow maps. We developed a multi-level decision tree algorithm, which obtained the optimal threshold values from the training data. Using Landsat-5 TM data as true values, we obtained the frequency distribution characteristics of each band from AVHRR data in the snow and non-snow areas at SR1, BT11, SR3/SR2, SR3-SR2, NDVI, and NDSI. The snow decision

tree includes three levels.

#### 1) First-level decision tree

SR1, BT11, and SR3/SR2, combined with DEM, were chosen as first-level discriminators. The first-level discriminators exclude specific non-snow pixels. Snow has high reflectance in the SR1 band and low BT in the thermal infrared BT11 band. Based on the frequency distributions of snow and non-snow

pixels for the first-level discriminators for Landsat-5 TM maps, a confidence level of 95% of snow samples was set to obtain the threshold value of possible snow and certain non-snow pixels. As shown in Table 5, for the samples before 2000, SR1 was >0.14 and BT11<274 K when DEM<1300 m, BT11 was <281 K when DEM≥1300 m, and SR3/SR2<0.50 were the possible snow images, while the



remaining pixels were non-snow pixels. The potential snow pixels were used as input for the second-

level decision tree.

**2)    Second-level decision tree**

The second-level decision tree was mainly used to obtain the snow information in the possible snow

pixels. The second-level discriminators were NDVI and SR3-SR2. Based on the frequency distributions

of possible snow pixels derived from the first-level decision tree, a confidence level of 99% of non-snow

samples was set to obtain the threshold value of certain snow and possible snow pixels. For the samples

before 2000, a pixel was classified as certain snow when NDVI < -0.16 and SR3-SR2 < -0.81 (Table 5).

Other pixels were considered the possible snow pixels, which were used as input for the third-level

decision tree.

**3)    Third-level decision tree**

NDSI was used as the third-level decision tree indicator due to its excellent discrimination ability of snow

cover and other land covers. Based on the frequency distributions of possible snow pixels derived from

the second-level decision tree, the optimal NDSI threshold value was calculated by a method similar to

that of the cloud test. Figure 4 shows the optimal NDSI scheme. Fig.4 (a) presents the NDSI frequency

distribution histogram of snow and non-snow pixels. The cross-point of snow and non-snow that has the

highest overall accuracy (85.87%) was chosen as the optimal NDSI threshold (0.73), as shown in Fig

4(b). The cross-point also represents a compromise for the snow omission (15.83%) and commission

error (13.03%). Thus, pixels with NDSI>0.73 were identified as snow for the samples before 2000.

Following the same strategy, optimal snow discrimination threshold values were obtained from AVHRR

data before and after 2000 (Table 5). Using the algorithm above, we first produced the AVHRR

preliminary SCE record for China based on the AVHRR SR V4.

**3.4 Gap-filling strategies**

For daily AVHRR preliminary SCE records, gaps due to frequent cloud obscuration or swath gaps

remained serious. Two gap-filling strategies described below were used to generate a spatially complete

daily AVHRR SCE record.





### 3.4.1 HMRF-based spatio-temporal modeling

Here, we present a spatio-temporal modeling technique for filling up gap pixels in daily snow cover estimates based on the time series of AVHRR preliminary SCE records. The spatio-temporal modeling technique integrated AVHRR preliminary SCE record spatial and temporal contextual information within a Hidden Markov Random Field (HMRF) model (Melgani and Serpico, 2003). Initially, Huang et al. (2018) utilized HMRF based spatio-temporal modeling for deriving and improving daily MODIS snow products. The core of this method is computing the total spatiotemporal energy for every gap from the neighborhood pixels and further classifying the gap pixels as snow pixels, non-snow pixels, or still gap pixels using

$$U_T(\beta_n N_{sp} N_{tp}) = \lambda_{st} U_{st}(\beta_n N_{sp} N_{tp}) \quad , \tag{3}$$

where $U_T$ is the total energy function of belonging to the class of $\beta_2$ ( $\beta_1$ denotes snow and $\beta_2$ denotes non-snow), $\lambda_{st}$ is the weight parameters, and $U_{st}$ is the spatio-temporal neighborhood cubic energy function. $N_{sp}$ $and$ $N_{tp}$ denote the spatial neighborhood and temporal neighborhood centered with the gap pixel, respectively.

Figure 5 illustrates our gap-filling process based on the HMRF technique. For a given gap at the center, we first calculated U($\beta_1$) and U($\beta_2$) based on a spatio-temporal, surrounding cube with 3 rows ×3 columns ×3 days. If U($\beta_1$) was > U($\beta_2$), gap pixels were classified as snow pixels. Otherwise, they were classified as non-snow pixels. If U($\beta_1$) = U($\beta_2$) or there were not sufficient valid pixels for calculating U($\beta_n$), we extended the spatio-temporal neighborhood to 3 rows ×3 columns ×5 days. If there were still insufficient valid pixels, the spatio-temporal neighborhood was expanded to 5 rows × 5 columns × 5 days. If the strategy above failed, gap pixels were maintained.

The HMRF-based modeling provided a rigorous interpolation framework for optimally integrating spatial-temporal contexts. To test the effect of HMRF-based interpolation for gap pixels, we used the monthly average gap ratio of the AVHRR preliminary SCE record from 1981 to 2019 before and after HMRF-based interpolation (table 6). The gap ratio of the AVHRR preliminary SCE record before HMRF-based interpolation was within 40% –60% (average: 47.8%), and the gap ratio after HMRF-based interpolation ranged between 0.2% and 6.4% (average: 2.7%). Almost 90% of gap pixels could be





reduced. The HMRF-based spatio-temporal model significantly improved the practicability of the AVHRR SCE product.

### 3.4.2 Interpolation based on passive microwave snow-depth data

Although most gap pixels were filled after interpolating the HMRF-based spatio-temporal model, there were still ~6% gaps left in the daily SCE data. Therefore, a fusion method combining the passive microwave daily snow-depth data and the AVHRR snow cover data was performed for these residual gap pixels. The passive microwave daily snow-depth data (25 km) were resampled to the same cell size as the AVHRR data (5 km) by the nearest neighbor interpolation method. If collocated snow depth was

$\geq$ 2-cm, the gap was considered a snow pixel. Otherwise, it was considered a non-snow pixel (Hao et al., 2019).

### 3.5 Postprocessing based on surface temperature and DEM

Because of their similar optical properties, ice-cloud pixels are sometimes mistaken for snow pixels, which will result in artifact snow covers in Southern China even during summers, where and when snow

is impossible. Referencing the MODIS algorithm, the postprocessing adopts LST products of ERA5 reanalysis and DEM to eliminate these snow pixels. The corresponding thresholds are given as below: the pixel is reclassified as snow-free when LST is $\geq$ 275 K, and DEM is $\leq$ 1300 km, or LST is $\geq$ 281 K, and DEM is $\geq$ 1300 km.

### 4 Accuracies of the NIEER AVHRR SCE product

**4.1 Methodology of accuracy evaluation**

A confusion matrix similar to that given in Table 7 is used to assess all associated AVHRR SCE data in the paper. Four kinds of accuracy metrics were used in this study followed on previous studies (Dong et al., 2014; Zhang et al., 2019), including the OA, the producer's accuracy (PA), the user's accuracy (UA), and Cohen's kappa (CK) value. The OA is the fraction of the correctly detected cases and all cases. The

PA measures the probability of correctly detected snow cases by AVHRR in the actual snow cases. The UA measures the proportion of true snow cases in all the detected snow cases by AVHRR. The sum of PA and omission error equals one, and the sum of UA and commission error equals one. (Arsenault et



al., 2014). CK value is an overall measurement of the agreement and is considered a more robust metric than OA (Cohen, 1960; Powers and Ailab, 2011).

**4.2 Validation with ground snow-depth measurements**

As mentioned above, we will use 38-year CMA ground snow-depth measurements at 191 stations to validate the new NIEER AVHRR SCE product. Table 8 presents an overview of validation results. The OA up to 87.4%. The value of PA (81.0%) was close to the UA (81.3%), which indicated that the algorithm sensibly performed a trade-off between the omission error (19.0%) and commission error (18.7%). In addition, the CK value was 0.717. According to the guidelines presented by Landis and Koch (1977), this would place the level of agreement as "substantial". All reveal on a whole the new NIEER AVHRR product is accurate and has a good agreement with measurements of CMA stations.

To validate the stability and reliability of the NIEER AVHRR SCE product, Fig.6 presents the four accuracy metrics' annual fluctuation over the past 38 years. The OA ranged within 80%–90%, the PA and UA ranged within 70%–90%, and the CK value ranged from 0.61 to 0.8. Several considerable annual fluctuations mainly occurred in 1993, 1994, and 2017, which were mainly caused by the poor quality of raw satellite data rather than the algorithm. In summary, the product maintained a higher precision with small annual fluctuations,which indicated the effectiveness and stability of the training framework with different thresholds before and after 2000.

Figure 7 further detailed accuracy metrics at each CMA station. From this figure, the OAs had higher values within 80%–90% in most stations across China, but PA, UA, and CK had low values with a clear spatial inconsistency. We found that the product performed well in North Xinjiang and the north of Northeast China where the stable snow was widely distributed. In contrast, the accuracy was relatively lower on the Qinghai-Tibet Plateau, Loess Plateau, in the Northeast of Inner Mongolia, and in the South of Northeast China, where snowpack may be instability due to patchy snow-cover features, rugged terrains, or rapid melt even in winter.

**4.3 Validation with Landsat-5 TM SCE maps**

The measurements from CMA stations can provide time-continuous validation. However, the "point to area" evaluation method ignores the spatial heterogeneity of satellite images within one pixel (Huang et al., 2011). The snow condition of an individual CMA station may not represent the larger area viewed



by AVHRR. The "area to area" method using higher-resolution images has pointed out a good way to assess snow spatial distribution of AVHRR SCE product.

In the study, 8 Landsat-5 snow maps were used to further evaluate the NIEER AVHRR product. Table 9 gives the validation results of our maps versus the Landsat-5 TM SCE maps. The OA was as high as 89.4%, the PA (90.2%) and UA (96.1%) were both greater than 90%, the omission error (9.8%) and commission error (3.9%) were both below 10%. The high UA and low PA revealed that the product has a slight tendency to underestimate the snow cover extent. The CK value (0.713) of the 'area to area' method also demonstrated 'substantial' agreement, which was close to that of ground measurements validation (0.717). Therefore, no matter from either point of view (ground measurements) or area of view (Landsat-5 SCE maps), the NIEER AVHRR product is accurate. In general,the new NIEER AVHRR SCE product is promising to better serve the climatic and other related studies in China.

Figure 8 further displays three intuitional examples demonstrating the detailed difference between the NIEER AVHRR SCE maps and the Landsat-5 SCE reference maps. The three images (serial number "C1, C3, and C6") were located in Northeast China, the Qinghai-Tibet Plateau, and North Xinjiang, respectively. It was clear that the NIEER AVHRR SCE maps agree much better with higher-resolution snow maps in a wide range of snow-covered areas. However, in the edges of snow-covered areas, the NIEER AVHRR SCE maps failed to identify most snow pixels in the Landsat-5 SCE maps, which could be explained by the low ability of our product to detect low fractional snow-covered pixels.

## 5 Discussion

### 5.1 Uncertainties of the NIEER AVHRR SCE product

The validation based on both CMA stations and Landsat TM images indicated that the NIEER AVHRR SCE product performs well for large and deep snow-covered areas. To explore the uncertainties of our product in the thin snow-covered areas, we set different snow depth (SD) thresholds based on CMA measurements to further evaluate the NIEER AVHRR SCE product. Figure 9 shows the accuracy metrics of the product under different SD thresholds (SD≥1 cm, SD≥2 cm, SD≥3 cm, SD≥4 cm, and SD≥5 cm). The results showed that the OA, UA, and CK values of the product decreased with increasing SD thresholds. While the PA values of the product increased with the increase of SD threshold. As SD increased, the UA presented a sharply decreasing trend and PA presented a slightly increasing trend. On





a whole, OA and CK values showed a significant decreasing trend. We can see our algorithm performed

well at lower SD thresholds,   which indicated the product has a better recognition ability for shallow

snow.

According to the snow cover temporal distribution feature in China, three seasonal snow periods were

defined, i.e., the snow accumulation period, stable snow period, and snow melting period. The snow

accumulation period is November. The stable snow period ranges from the beginning of December of

the year to the end of February, and the snow melting period is March. Figure 10 presents the accuracy

results of the NIEER AVHRR SCE product in different snow periods. The OAs of the accumulation

period (87.7%), stable period (86.7%) and melting period (89.0%) showed a similar response. However,

the PAs, UAs and CK values of the accumulation and melting periods were markedly lower than those

of the stable snow period. The product had the highest omission errors (29.5%) during the accumulation

period because of the mixed pixels in the early snowfall seasons; while the product had the highest

commission error (30.3%) during the melting period due to the influence of wet snow.

**5.2 Comparison of NIEER AVHRR and JASMES SCE product**

To more objectively assess our product, we compared the NIEER AVHRR SCE product with JASMES

SCE products. Since the JASMES SCE product was only generated by AVHRR data from 1981 to 1999,

comparisons were made against the same ground snow-depth reference measurements in 19 snow

seasons (1981-1999). Table 10 lists the comparison of the accuracy metrics. Our products performed

well, with OA, PA, UA, and CK values of 86.1%, 80.3%, 78.7%, and 0.690, respectively. The JASMES

SCE products performed poorly, with total OA, PA, UA, and CK values amounting to 71.8%, 39.2%,

68.1%, and 0.321, respectively. It means that our product clearly outperforms the JASMES product.

Relative to the JASMES SCE product, the NIEER AVHRR OA increased approximately 15 percent, the

omission error dropped from nearly 60% to 19.7%, the commission error dropped from 31.9% to 21.3%,

and the CK value increased by more than 114%. The JASMES product markedly underestimated the

snow in China. In addition, there were about 50 thousand validation samples in our product and only

about 36 thousand SD measurements in that of the JASMES product. Thus, our product should fill more

gap pixels than JASMES. On the whole, the snow and cloud detection algorithm and the gaps-filled

strategy of our product performed better than those of JASMES.



To better figure out the spatial distribution difference between the two sets of products, comparison maps were constructed for November 15, 1985. Figure 11 presents the two SCE maps and their difference. There were significant differences in mapped snow extent between the two maps in the three major

seasonal snow regions in China, i.e., North Xinjiang, Northeast China, and the Qinghai-Tibet Plateau. Our product mapped more snow in North Xinjiang, the Qinghai-Tibet Plateau, and the non-forest area in the Northeast of China than JASMES. The most considerable discrepancy occurred on the Qinghai-Tibet Plateau, where our product identified more snow-covered areas than JASMES. JASMES maps had more snow in the forested area of Northeast China than our product. Three improvements contributed to this

phenomenon. Firstly, the snow algorithm proposed improved snow discrimination accuracy and reduced omission errors largely. Secondly, the cloud detection algorithm effectively improved the cloud-snow confusion, which identified the snow pixels that were misidentified as clouds pixels in the JASMES. Thirdly, the gaps-filled strategy provided complete spatial coverage of snow cover.

## 6 Data availability

The NIEER AVHRR SCE product was named in a manner of NIEER_GF AVHRR SCE_yyyymmdd_DAILY_5km_V01 (V01 denotes the first version). It has a spatial resolution of 5 km and a daily temporal resolution. It spans latitude 16-56 °N and longitude 72-142 °E, and now is freely accessible at https://dx.doi.org/10.11888/Snow.tpdc.271381 (Hao et al., 2021). Detailed information on the product is listed in Table 11. The values in the product are classified as non-snow (0), water (4),

filling value (255), snow from AVHRR (1), snow from HMRF (2), and snow from SD (3).

## 7 Conclusions

This study developed a multi-level decision tree algorithm for cloud and snow discrimination based on AVHRR SR V4 data in China. We used an improved gap-filling technique to fill in the missing values

of the product and generate a daily NIEER AVHRR SCE product with a spatial resolution of 5 km across China from 1981 to 2019. The AVHRR SCE product was validated using snow depth measurements provided by the China Meteorological Administration and higher spatial resolution SCE maps derived from Landsat-5 TM.

The OA of the NIEER AVHRR product was 87.4%, a high accuracy, while the PA and UA were 81.0% and 81.3%, respectively. The PA and UA were similar, showing that the algorithm of the NIEER AVHRRA product performed a trade-off between commission and omission errors. The CK value was 0.717, which indicated that the product had an agreement level of "Substantial". Considering the limitations of point-to-area validation, the overall OA, PA, UA, and CK values were 89.4%, 90.2%, 96.1%, and 0.713, respectively, using Landsat-5 TM area-to-area, which showed the same trend of

accuracy as the point validation. Therefore, no matter from either point of view or area of view, our AVHRR SCE product has high accuracy.

The performance of the NIEER AVHRR product in China was compared with the existing JASMES AVHRR SCE product. The OA, PA, UA, and CK value of the NIEER product were 86.1%, 80.3%, 78.7%, and 0.690, and those of JASMES were 71.8%, 39.2%, 68.1%, and 0.321. Compared with the

JASMES product, the NIEER product OA increased approximately 15 percent, the omission error dropped from nearly 60% to 19.7%, the commission error dropped from 31.9% to 21.3%, and the CK value increased by more than 114%. Accordingly, the NIEER AVHRR product had a higher accuracy than the JASMES product. Furthermore, the NIEER product provides a completely gap-free product for China, permitting its wide applications.

Finally, we assessed the behavior of the NIEER AVHRR product during the snow accumulation, stable snow, and melting periods. The SCE performed best during the stable period, and the product was more accurate in the snow accumulation than the melting period. In general, the algorithm had a relatively high ability to identify shallower snow, but some uncertainties existed in patchy snow areas, regarding thinner snow, and in rugged terrain areas. As a long-term record, the dataset will provide a valuable data source

for analyzing the influence of climate changes on the cryosphere on multiple time scales.

**Author contribution.**

XH and GH designed the study and developed the methodology; XH wrote the manuscript; TC, JW, QZ, HL, QY revised the manuscript. WJ, XS and HZ developed the python code.

**Competing interests.**

The authors declare that they have no conflict of interest.



**ACKNOWLEDGEMENTS**

The authors would like to thank the China Meteorological Administration for ground snow-depth measurements, the National Oceanic and Atmospheric Administration (NOAA) and the Japan Aerospace Exploration Agency (JAXA) for satellite data. We also acknowledge that the Google Earth Engine

dramatically facilitated the work on image re-processing.

**Financial support.**

This research was supported by the Strategic Priority Research Program of the Chinese Academy of Sciences (Grant No. XDA19070101), the National Natural Science Foundation of China (Grant No. 41971325, 41971399, 41801283), the Science & Technology Basic Resources Investigation Program of

China (Grant No. 2017FY100502), the National Key Research and Development Program of China (Grant No. 2019YFC1510503).

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



**Table 1: The details of spectral bands from CDR of AVHRR Surface Reflectance (Version 4) from GEE platform.**

| GEE Band | Abbreviation | Wavelength (μm) | Description |
| --- | --- | --- | --- |
| SREFL_CH1 | SR1 | 0.58-0.68 | Surface Reflectance at 0.64um |
| SREFL_CH2 | SR2 | 0.725-1.00 | Surface Reflectance at 0.86um |
| SREFL_CH3 | SR3 | 3.55-3.93 | Surface Reflectance at 3.75um |
| BT_CH3 | BT37 | 3.55-3.93 | Brightness temperature at 3.75um |
| BT_CH4 | BT11 | 10.30-11.30 | Brightness temperature at 11.0um |
| BT_CH6 | BT12 | 11.50-12.50 | Brightness temperature at 12.0um |



**Table 2: The number of training scenes using Landsat-5 TM**

| Type of sample | Number of Landsat-5 TM scenes | Year |
|---|---|---|
| Snow samples | 1293 | Before 2000 |
| | 6695 | After 2000 |
| Non-snow samples | 1670 | Before 2000 |
| | 5774 | After 2000 |
| Cloud samples | 79 | Before 2000 |
| | 125 | After 2000 |

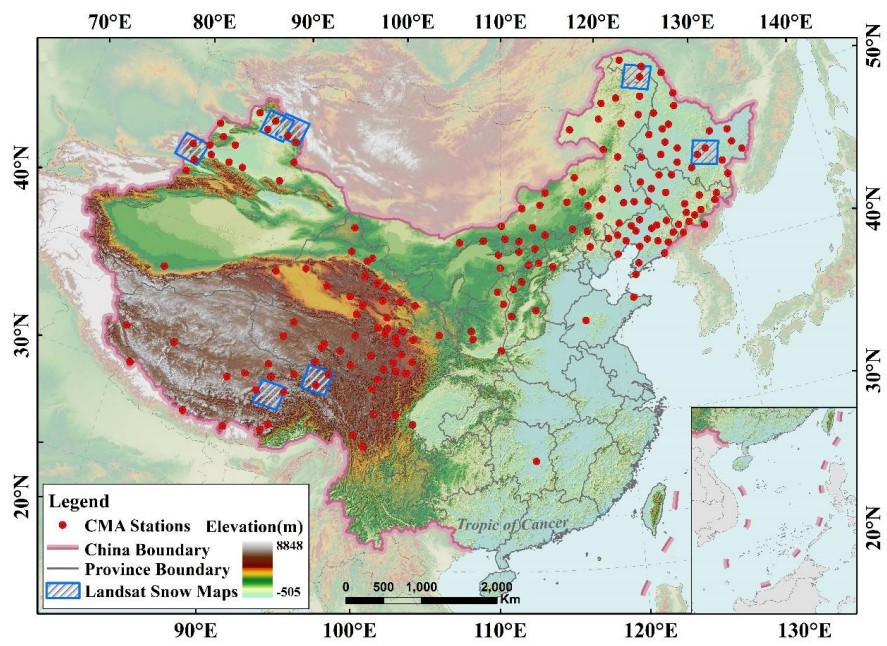


**Figure 1: The geographic location of study area and the spatial distribution of climate stations and Lansat-5 validation dataset. The elevation data were derived from Shuttle Radar Topography Mission (SRTM).**



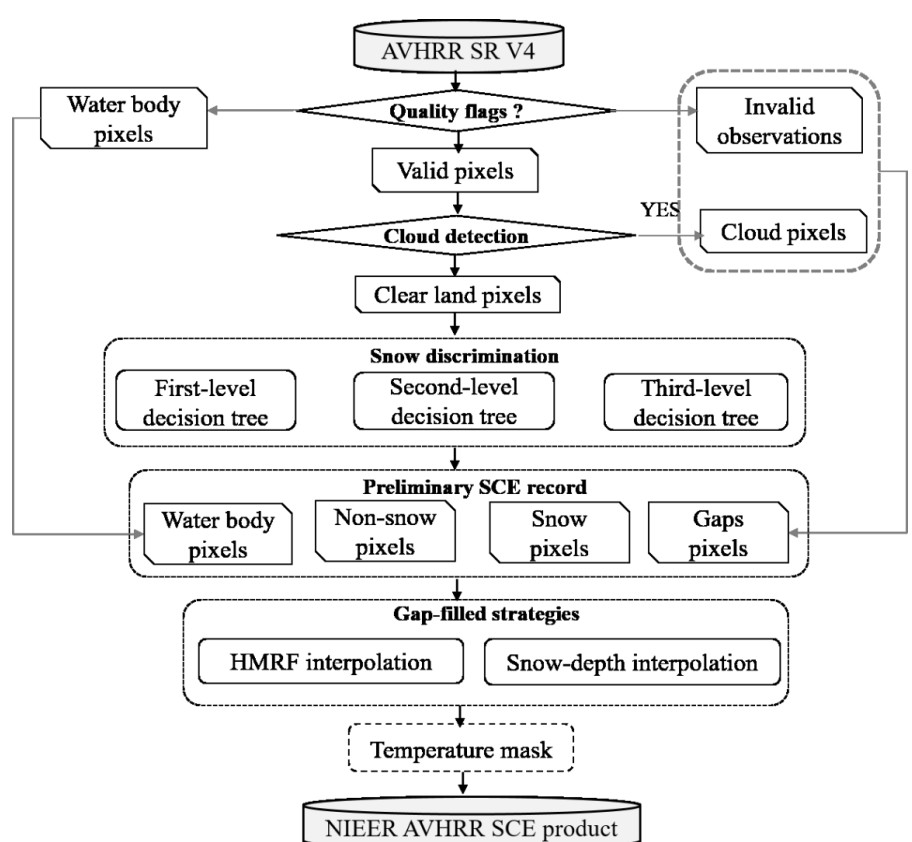

**Figure 2: Generation flowchart of NIEER AVHRR snow cover extent product (NIEER AVHRR SCE)**



**Table 3: The descriptions of quality control of AVHRR SR V4**

| Bitmask | Description | Value = 0 | Value = 1 |
|---|---|---|---|
| 15 | Polar flag (latitude over 60 degrees (land) or 50 degrees (ocean)) | No | Yes |
| 14 | BRDF-correction issues | No | Yes |
| 13 | RHO3 value is invalid | No | Yes |
| 12 | Channel 5 value is invalid | No | Yes |
| 11 | Channel 4 value is invalid | No | Yes |
| 10 | Channel 3 value is invalid | No | Yes |
| 9 | Channel 2 value is invalid | No | Yes |
| 8 | Channel 1 value is invalid | No | Yes |
| 7 | Channel 1-5 are valid | No | Yes |
| 6 | Pixel is at night (height solar zenith) | No | Yes |
| 5 | Pixel is over dense dark vegetation | No | Yes |
| 4 | Pixel is over sunglint | No | Yes |
| 3 | Pixel is over water | No | Yes |
| 2 | Pixel contains cloud shadow | No | Yes |
| 1 | Pixel is cloudy | No | Yes |
| 0 | Unused | No | Yes |




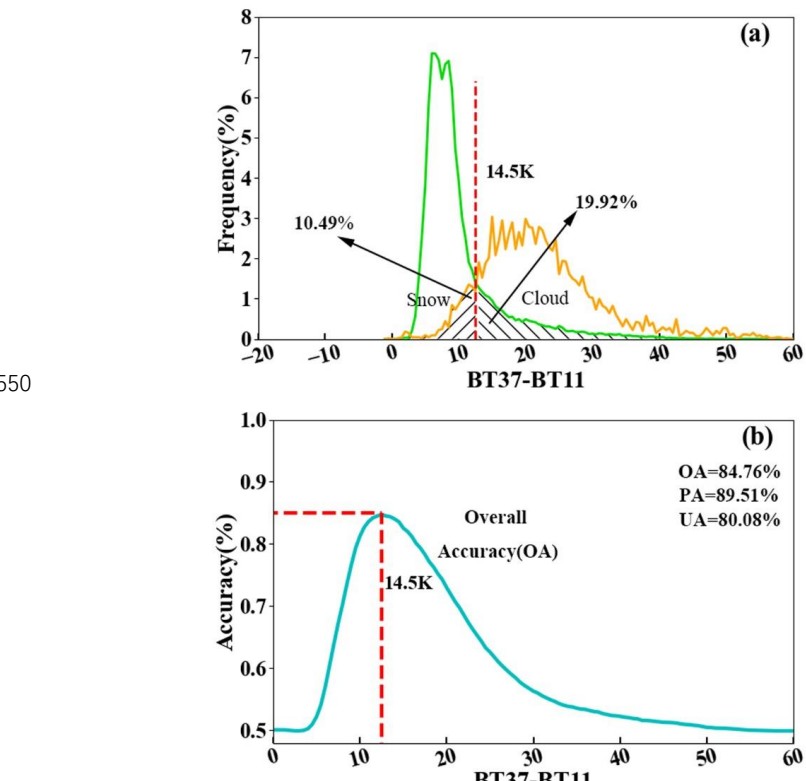

**Figure 3: The frequency distribution of BT37-BT11 and optimal threshold acquisition of snow and cloud from A1 before 2000. Figure 3(a) shows the frequency distribution of snow and cloud on AVHRR, and Figure 3(b) shows the determination of optimal threshold for cloud detection.**






**Table 4. Cloud detection tests and the corresponding thresholds. Target A indicates high and cold land (elevation > 300m and BT11 < 260 K), which have four types: A1~A4; Target B indicates the remaining land, which includes ten types: B11~B10. The cloud detection test was conducted from the top of the list to the bottom for each target. If the switch of the cloudy flag was "on", the pixel was set to cloudy when the threshold tests met the conditions listed on the right-hand side. If the switch was "off", the pixel identified as cloudy in the previous tests was reset to clear.**

| Target | Target serial number | Switch | Elevation (m) | SR1 | SR2 | SR3 | SR1-SR2 | NDVI | BT11(K) | Before 2000 BT37-BT11(K) | After 2000 BT37-BT11(K) | BT11-BT12(K) |
|---|---|---|---|---|---|---|---|---|---|---|---|---|
| A: High or cold land DEM>300 and BT11<260K) | A1 | On | <3000 | | | | | | ≥240 | >14.5 | >19.5 | |
| | A2 | On | ≥3000 | | | | | | ≥240 | >15.5 | >20 | |
| | A3 | On | | | | | | | <240 | >21.0 | >31 | |
| | A4 | On | | | | >0.1 | >0.02 | | | >25.5 | >33.5 | |
| | B1 | On | | | | | | | <260 | >14 | >16 | |
| | B2 | On | | | | | >-0.02 | | <310 | >10.5 | >16.5 | |
| | B3 | On | | >0.3 | | | >-0.02 | | <293 | >11.5 | >17.5 | |
| B : Other land DEM<300 or BT11>=260K | B4 | On | | | >0.4 | | >-0.03 | | <293 | >11.5 | >18.0 | >-1 |
| | B5 | On | | | >0.4 | | | | <278 | >11.5 | >19.5 | >-1 |
| | B6 | On | | >0.3 | | >0.02 | | | | >11.5 | >18 | |
| | B7 | Off | | | | | | >0.5 | >288 | | | |
| | B8 | Off | | | | | | | >310 | | | |
| | B9 | Off | >1000 | <0.4 | | | <-0.04 | | >275 | | | |
| | B10 | Off | | | | | <-0.04 | | >300 | | | |

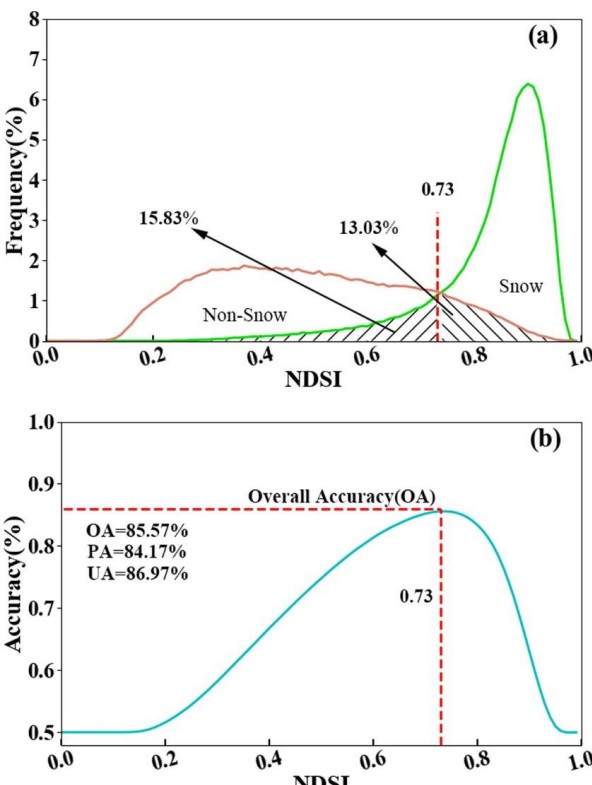

**Figure 4: NDSI frequency distribution histogram and optimal threshold acquisition of snow and non-snow before 2000. (a) is the frequency distribution of snow and non-snow on AVHRR, and (b) is the optimal NDSI threshold value.**



**Table 5. Snow discrimination algorithm and its threshold values.**

| Target | Snow | SR1 | BT11(K) | Elevation (m) | SR3/SR2 | SR3-SR2 | NDVI | NDSI |
|--------|------|-----|---------|---------------|---------|---------|------|------|
| A: Before 2000 | Snow1 | >0.14 | <274 | <1300 | <0.5 | <-0.81 | | |
| | | >0.14 | <281 | ≥1300 | <0.5 | <-0.81 | | |
| | Snow2 | >0.14 | <274 | <1300 | <0.5 | | <-0.16 | |
| | | >0.14 | <281 | ≥1300 | <0.5 | | <-0.16 | |
| | Snow3 | >0.14 | <274 | <1300 | <0.5 | ≥-0.81 | ≥-0.16 | >0.73 |
| | | >0.14 | <281 | ≥1300 | <0.5 | ≥-0.81 | ≥-0.16 | >0.73 |
| B: After 2000 | Snow1 | >0.14 | <275 | <1300 | <0.56 | <-0.77 | | |
| | | >0.14 | <281 | ≥1300 | <0.56 | <-0.77 | | |
| | Snow2 | >0.14 | <275 | <1300 | <0.56 | | <-0.05 | |
| | | >0.14 | <281 | ≥1300 | <0.56 | | <-0.05 | |
| | Snow3 | >0.14 | <275 | <1300 | <0.56 | ≥-0.77 | ≥-0.05 | >0.65 |
| | | >0.14 | <281 | ≥1300 | <0.56 | ≥-0.77 | ≥-0.05 | >0.65 |


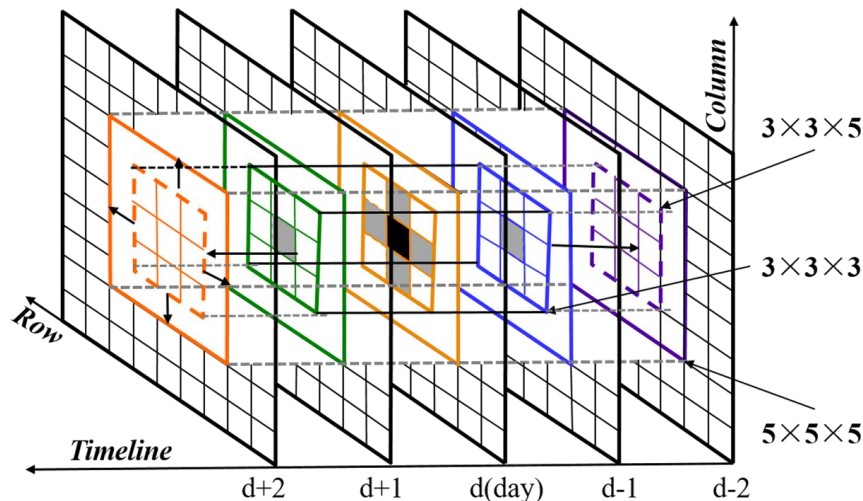

**Figure 5: Diagram of the HMRF-based gap-filling process used in the study.**





**Table 6. The monthly average gap ratio of AVHRR preliminary SCE record in China before and after HMRF-based spatio-temporal interpolation from 1981 to 2019.**

| Month | Gap ratio before interpolation (%) | Gap ratio after interpolation of HMRF (%) |
|---|---|---|
| 1 | 51.4 | 2.0 |
| 2 | 55.2 | 2.7 |
| 3 | 57.0 | 2.5 |
| 4 | 52.1 | 0.9 |
| 5 | 50.3 | 1.0 |
| 6 | 48.1 | 0.8 |
| 7 | 46.0 | 1.3 |
| 8 | 40.1 | 0.2 |
| 9 | 39.5 | 2.4 |
| 10 | 39.8 | 5.6 |
| 11 | 44.0 | 6.0 |
| 12 | 49.6 | 6.4 |
| Average | 47.8 | 2.7 |

**Table 7 Description of a confusion matrix of snow classification between NIEER AVHRR SCE product and truth value that reference ground snow-depth measurements or Landsat-5 TM SCE maps.**

|  |  |  | NIEER AVHRR SCE product | |
| --- | --- | --- | --- | --- |
|  |  |  | Snow | Non-snow |
| Ground | snow- | Snow | SS | SN |
| depth |  | Non-snow | NS | NN |

Overall Accuracy (OA)
$$OA = \frac{SS + NN}{T}$$

Producer's Accuracy (PA)
$$PA = \frac{SS}{SS + SN}$$

User's Accuracy (UA)
$$UA = \frac{SS}{SS + NS}$$

Cohen's Kappa coefficient (CK).
$$CK = \frac{OA - P}{1 - P}$$

Where, $T = SS + SN + NS + NN$

$$P = \left(\frac{SS + NS}{T} \times \frac{SS + SN}{T}\right) + \left(\frac{NN + NS}{T} \times \frac{NN + SN}{T}\right)$$

Note: SS, SN, NS and NN are all numbers. SS reps the number of cases that AVHRR predicts Snow and the ground snow-depth measures Snow. SS reps the number of cases that AVHRR predicts Non-snow and the ground snow-depth measures Non-snow. SN reps the number of cases that AVHRR predicts Non-snow while the ground snow-depth measures snow. NS reps the number of cases that AVHRR predicts Snow while the ground snow-depth measures Non-snow.




**Table 8 A confusion matrix for NIEER AVHRR SCE maps versus ground snow-depth measurements**

| | Class | NIEER AVHRR SCE | |
| --- | --- | --- | --- |
| | | Snow | Non-snow |
| Ground snow-depth measurements | Snow | 282239 | 66167 |
| | Non-snow | 64759 | 622381 |
| | OA | 87.4% | |
| | PA | 81.0% | |
| | UA | 81.3% | |
| | CK | 0.717 | |

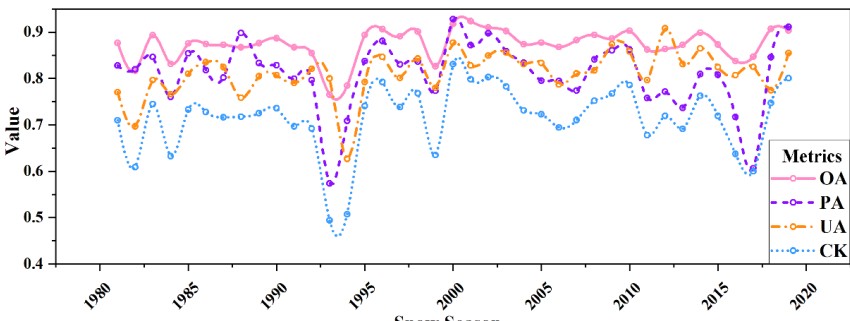

**Figure 6: Accuracy fluctuations of NIEER AVHRR product base on ground snow-depth measurements in the**

**past 38 years.**

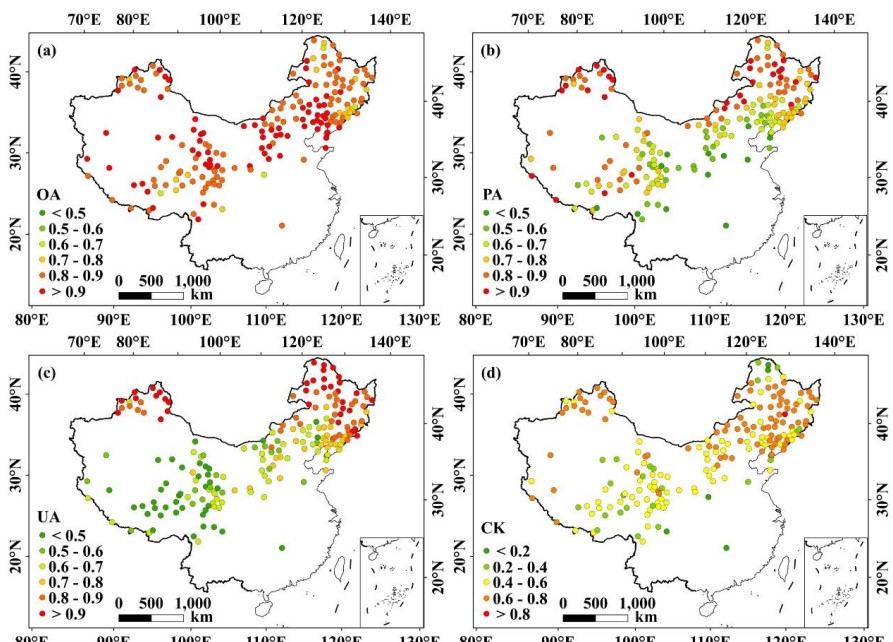

**Figure 7: Point-based accuracy results of NIEER AVHRR product: (a) OA; (b) PA; (c) UA; (d) CK. The snow depth of 191 climate stations used is provided by the China Meteorological Administration (CMA). OA, PA, UA and CK represent overall accuracy, producer's accuracy, user's accuracy, and Cohen's Kappa coefficient.**




**Table 9 The accuracy of NIEER AVHRR SCE maps versus Landsat-5 TM SCE maps. C1~C8 denotes the different Landsat-5 TM SCE.**

| Path/row | Serial number | Date | Cloud percentage | Snow percentage | OA | PA | UA | CK |
|---|---|---|---|---|---|---|---|---|
| 116028 | C1 | 19970312 | 2.0% | 77.2% | 87.9% | 88.3% | 95.9% | 0.678 |
| 121024 | C2 | 20160319 | 1.8% | 96.4% | 98.1% | 100.0% | 98.1% | 1 |
| 135038 | C3 | 19961109 | 1.0% | 66.5% | 79.5% | 81.0% | 87.9% | 0.552 |
| 137039 | C4 | 19961123 | 2.0% | 50.7% | 78.2% | 65.7% | 88.5% | 0.566 |
| 142027 | C5 | 19870323 | 0.0% | 96.1% | 97.2% | 100.0% | 97.2% | 0.036 |
| 143027 | C6 | 20051110 | 2.0% | 48.6% | 93.1% | 86.7% | 99.8% | 0.863 |
| 147029 | C7 | 20160222 | 1.1% | 89.0% | 90.6% | 91.4% | 98.0% | 0.587 |
| 147029 | C8 | 19970217 | 2.0% | 88.3% | 89.8% | 90.9% | 97.7% | 0.560 |
| | Total | | | | 89.4% | 90.2% | 96.1% | 0.713 |


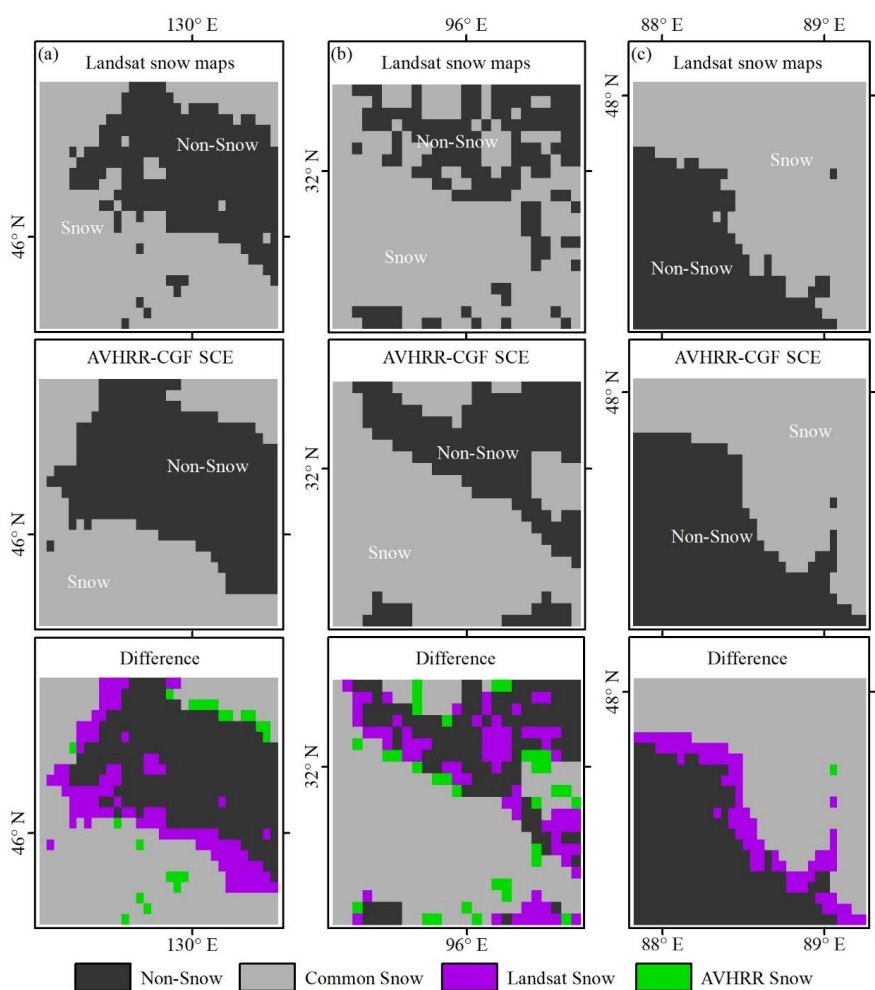

**Figure 8: Comparison of Landsat reference image with NIEER AVHRR SCE images. (a) is located in Northeast China on Mar. 12st, 1997;(b) is located in Qinghai-Tibet Plateau on Nov. 9st, 1996; (c) is located in North Xinjiang on Nov. 10st, 2005.**


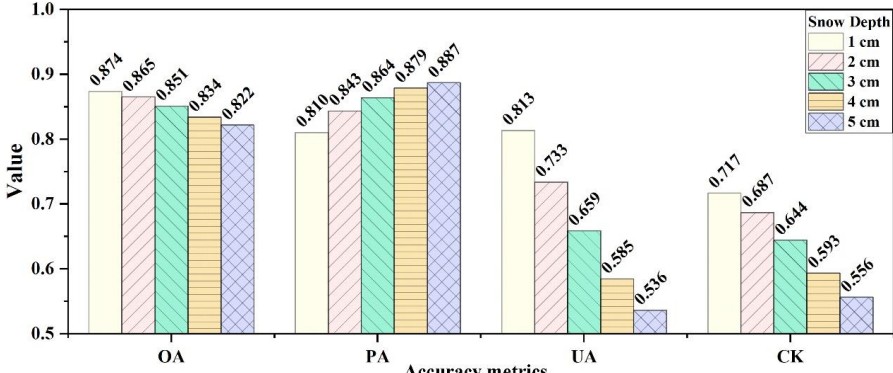

**Figure 9: Histogram of accuracy results of NIEER AVHRR SCE product under different snow depth thresholds.**


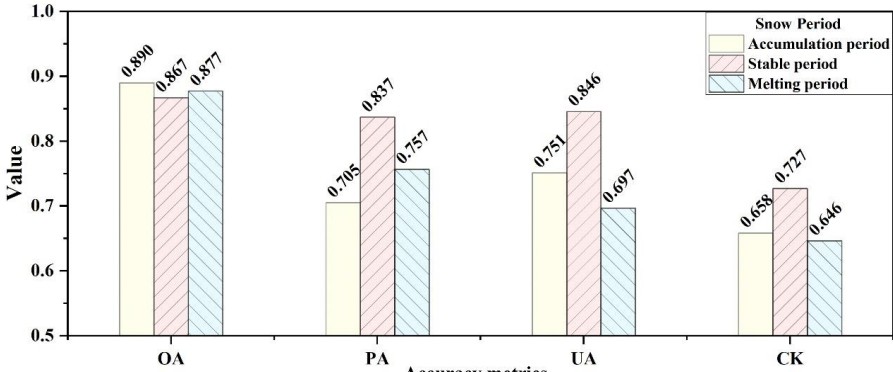

**Figure 10: Histogram of accuracy results of NIEER AVHRR SCE product in different snow periods, including accumulation period, stable period, melting period**






**Table 10 The confusion matrix and accuracy results of NIEER AVHRR and JASMES SCE product based on snow depth measurements from CMA. OA, PA, UA and CK.**

|  |  | NIEER AVHRR SCE | | JASMES SCE | |
|---|---|---|---|---|---|
|  | Class | Snow | Non-snow | Snow | Non-snow |
| Ground snow-depth measurements | Snow | 134260 | 32946 | 50335 | 78148 |
|  | Non-snow | 36367 | 295890 | 23594 | 209149 |
| OA |  | 86.1% | | 71.8% | |
| PA |  | 80.3% | | 39.2% | |
| UA |  | 78.7% | | 68.1% | |
| CK |  | 0.690 | | 0.321 | |

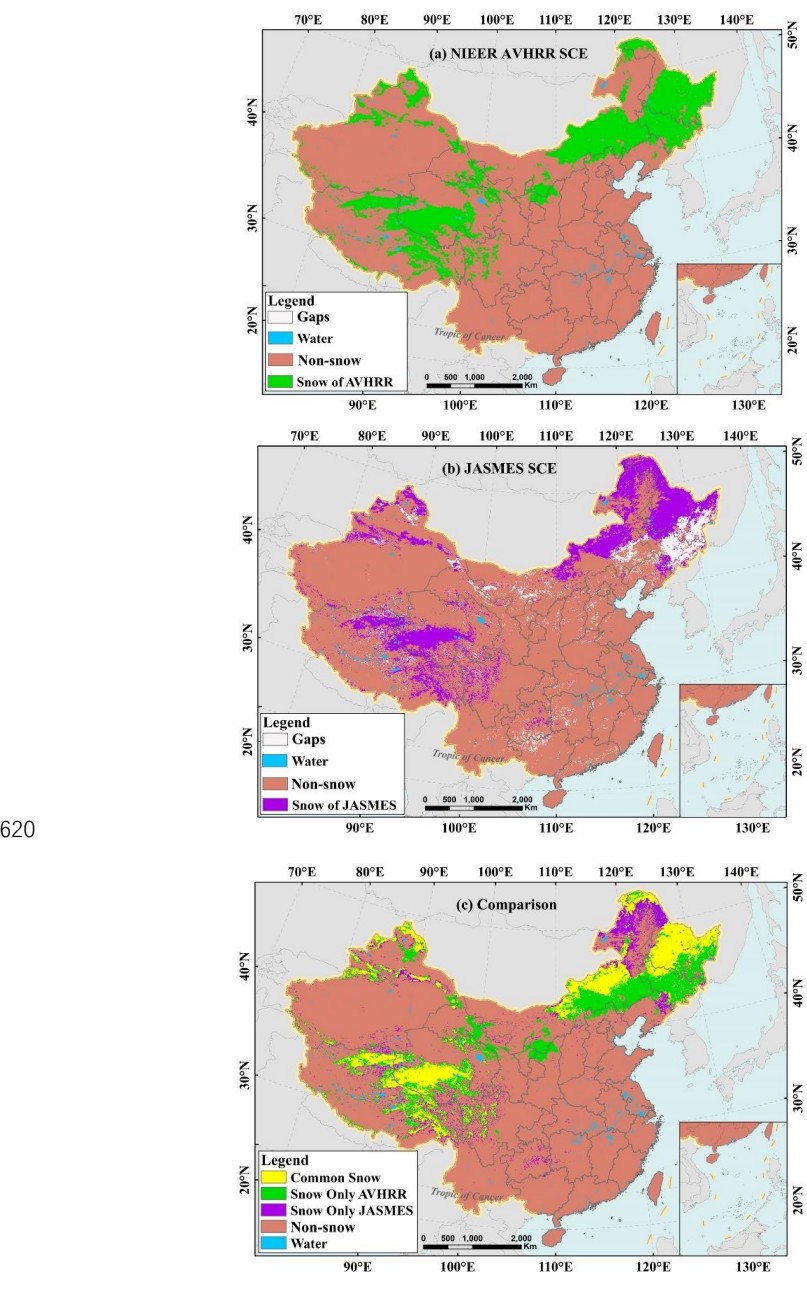

**Figure 11: Comparison of snow cover maps between the NIEER AVHRR and JASMES SCE map on November 15, 1985. (a) is NIEER AVHRR SCE map; (b) is JASMES SCE map; (c) comparison between the two snow maps.**





**Table 11 The description of NIEER AVHRR SCE product**

| Classification | values | Description |
| --- | --- | --- |
| Snow | 1 | Snow from AVHRR |
|  | 2 | Snow from HMRF |
|  | 3 | Snow from SD |
| Non-snow | 0 | Non-Snow form AVHRR |
| Water | 4 |  |
| Filling value | 255 | Filling value |