# Peer review of "The NIEER AVHRR snow cover extent product over China – A long-term daily snow record for regional climate research"

_Earth System Science Data, 2021_

## Community Comment (CC1)

Responses to Referee 1:

Thank you for your positive comments and helpful suggestions. This document provides point-by-point answers to your remarks. We have worked in particular on:

• describing the discrimination algorithm of snow and cloud cover clearly.

• emphasizing the dataset and weakening the methodology in the Conclusion

In the document, lines in **bold** echo your comments for ease of reading, the revised manuscript

with new elements in green.

We sincerely hope that these corrections can reach your expectations.

**This paper proposes an a long-term AVHRR snow cover extent product from 1981 until 2019 over China. The product has the spatial resolution of 5-km and the daily temporal resolution, and is a completely gap-free product, which is produced through quality control, cloud detection, snow discrimination, and gap-filling. The validations based on ground measurement and Landsat-5 snow maps both demonstrate its higher accuracy than that of the JASMES AVHRR product. As a long-term record, the dataset will provide a valuable data source for analyzing the influence of climate changes on the cryosphere on multiple time scales. The need for such a dataset is well justified and the authors cite ample relevant literature. The paper is basically well-written and presented.**

**1. it is best to delete "Using the Google Earth Engine (GEE) platform" in the first sentence because GEE is just the platform of producing the product, it is not the main contribution of the study. However, I suggest to add a subsection in the section of "data and preprocessing" to describe the computing platform and the reason of choosing GEE.**

**Response**: Thanks for this helpful suggestion. We have deleted "Using the Google Earth Engine (GEE) platform" in the first sentence, described the advantages of the GEE platform, and explained why the GEE platform is used.

A long-term AVHRR snow cover extent (SCE) product from 1981 until 2019 over China has been generated by the snow research team in the Northwest Institute of Eco-Environment and Resources (NIEER), Chinese Academy of Sciences.

Google established the Google Earth Engine (GEE) cloud computing platform from 2012. GEE enables academics to quickly access massive amounts of remote sensing data without downloading it, which could support scientific analysis and visualization of geospatial datasets with petabyte-scale (Gorelick,2012). In this study, all AVHRR SR V4 images were processed by GEE cloud platform.

New reference: Gorelick, N.: Google Earth Engine. Gebruiker Woody Bousson/kladblok, 2012.

**2. why not add the dataset of 2020 year?**

**Response:** Thanks for your suggestion. In the study, the NOAA Climate Data Record (CDR) of AVHRR Surface Reflectance Version 4 (AVHRR SR V4) was used as primary input data. The AVHHR CDR V4 used in our work is available from 1981-06-24T 00:00:00 to 2019-05-16T 00:00:00. Our work also indicated that the low quality of AVHHR after 2018 has a noticeable impact on the accuracy of snow cover extent products. This dataset developed mainly focused on the data before 2000. After 2019, the AVHRR CDR V4 product will no longer be produced due to satellite data quality issues. To extend the time series, we prefer to choose newly launched satellite sensors, like MODIS or NPP products.

**3. Is it sufficient enough to validate so big product using just eight Landsat-5 images?**

**Response:** We added the description of the three major snow-covered regions (Northern Xinjiang, Northeast China, and Qinghai-Tibet Plateau) of China in figure 1. In addition, we added two new validation images in Northeast China, and removed a similar validation image. A total of 9 Landsat-5 images distributed in the three major snow-covered regions were selected as validation data. And the detailed information of each verification image is shown in new table 9 (All serial number of images are rearranged). The main factors for snow cover detection are the type of land cover and topography. We chose three sceneries in Northern Xinjiang, one scene is located in the flat area of Altay (C8), one scene is in the Altai Mountains (C7), and the other scene is in the Tianshan Mountains (C9). The Northeast China, snow area is relatively flat. The snow is mainly distributed in forest areas, cultivated land and grassland, so we choose two forest area images (C1, C3), located

in Greater Khingan Range area and Small Khingan Range area, respectively, with one farmland (C2) and grassland in Inner Mongolia (C4). The snow cover of the Qinghai-Tibet Plateau is mainly distributed in mountainous areas. And the verification data we selected are mainly distributed in mountainous areas (C5, C6). The main types of land cover are grassland and bare land. Considering the above factors, a total of 9 Landsat-5 validation images are representative and can meet the requirements.

[Figure]

Figure 1: The geographic location of study area and the spatial distribution of major snow-covered regions, climate stations and Lansat-5 validation dataset. The elevation data were derived from Shuttle Radar Topography Mission (SRTM).

Another validation with reference to higher-resolution snow maps derived from Landsat-5 Thematic Mapper (TM) images demonstrates an overall accuracy of 87.3%, a producer's accuracy of 86.7%, a user's accuracy of 95.7%, and a Cohen's kappa value of 0.695.

A total of 9 Landsat-5 TM snow maps were used as the validation dataset (Fig.1). The training and validating samples were evenly distributed across China's major seasonally snow-covered regions (including North Xinjiang, Northeast China, and the Qinghai-Tibet Plateau) to ensure reliability and representativeness.

In the study, 9 Landsat-5 snow maps were used to further evaluate the NIEER AVHRR product. Table

9 gives the validation results of our maps versus the Landsat-5 TM SCE maps. The OA was as high as 87.3%. The high UA and low PA revealed that the product has a slight tendency to underestimate the snow cover extent. The CK value (0.695) of the 'area to area' method also demonstrated 'substantial' agreement, which was close to that of ground measurements validation (0.717).

Figure 8 further displays three intuitional examples demonstrating the detailed difference between NIEER AVHRR SCE maps and Landsat-5 SCE reference maps. The three images (serial number "C1, C5, and C8") were located in Northeast China, the Qinghai-Tibet Plateau, and North Xinjiang, respectively.

Considering the limitations of point-to-area validation, the overall OA, PA, UA, and CK values were 87.3%, 86.7%, 95.7%, and 0.695, respectively, using Landsat-5 TM area-to-area, which showed the same trend of accuracy as the point validation.

**Table 9 The accuracy of NIEER AVHRR SCE maps versus Landsat-5 TM SCE maps. C1~C9 denotes the different Landsat-5 TM SCE.**

| Path/row | Serial number | Date | Cloud percentage | Snow percentage | OA | PA | UA | CK |
|----------|---------------|------|------------------|-----------------|------|------|------|------|
| 116028 | C1 | 19970312 | 2.0% | 77.2% | 87.9% | 88.3% | 95.9% | 0.678 |
| 118029 | C2 | 20161109 | 0.2% | 88.0% | 84.5% | 82.8% | 99.5% | 0.519 |
| 121024 | C3 | 20160319 | 2.0% | 96.4% | 98.1% | 100.0% | 98.1% | 1 |
| 127031 | C4 | 20180130 | 1.1% | 45.3% | 82.0% | 63.0% | 96.1% | 0.626 |
| 135038 | C5 | 19961109 | 1.0% | 66.5% | 79.5% | 81.0% | 87.9% | 0.552 |
| 137039 | C6 | 19961123 | 2.0% | 50.7% | 78.2% | 65.7% | 88.5% | 0.566 |
| 142027 | C7 | 19870323 | 0.0% | 96.1% | 97.2% | 100.0% | 97.2% | 0.036 |
| 143027 | C8 | 20051110 | 2.0% | 48.6% | 93.1% | 86.7% | 99.8% | 0.863 |
| 147029 | C9 | 20160222 | 1.1% | 89.0% | 90.6% | 91.4% | 98.0% | 0.587 |
| | Total | | | | 87.3% | 86.7% | 95.7% | 0.695 |

**4.Line 17, NIEER is the abbreviation of the authors institute, "the new NIEER product" is difficult to project to the new AVHRR snow cover extent product, because many products are produced by the institute.**

**Response:** Thanks for your helpful suggestion. Referring to NASA NSIDC MODIS series, we plan to publish some snow products, including MODIS SCE, AVHRR SCE, and MODIS FSC. We revised "the new NIEER product" by "the NIEER AVHHR SCE products" to distinguish from the other products.

The NIEER AVHHR SCE product has the spatial resolution of 5-km and the daily temporal resolution, and is a completely gap-free product, which is produced through a series of processes such as the quality control, cloud detection, snow discrimination and gap-filling (GF).

**5.Line 27, "nearly 40%" should be changed to precise number.**

**Response:** Thanks for the careful check. According to our validation results, we have changed "nearly 40%" into "60.8%".

For example, the overall accuracy of our products was 15% higher than the well-known JAMES AVHRR product. The omission error dropped from 60.8% to 19.7%, the commission error dropped from 31.9% to 21.3%, and the CK value increased by more than 114 percent.

**6.Line 37, "these decades" refers to which decades? The citing references was published in 2005 and 2018.**

**Response:** Thanks for your helpful suggestion. We have replaced "over these decades" with "over the past several decades". Generally speaking, over these days, years or decades indicate during the past several days, years or decades.

With the continuous warming of the global climate, snow cover on the Earth has been shrinking evidently over the past several decades (Barnett et al., 2005; Bormann et al., 2018).

**7.Line 38, "is not only of particular importance for climate research but also an indispensable**

**indicator of climate change" needs to be rephrased.**

**Response:** Thanks for your suggestion. We improved this sentence, as marked as red in lines 38-39.

Therefore, long-term snow cover data are particularly important for climate research and are also a crucial indicator of climate change.

**8.Line 56, the abbreviation SCE has been described in the last paragraph.**

**Response:** Thanks for the helpful suggestion, and we deleted it.

The Japan Aerospace Exploration Agency (JAXA) recently issued the long-term SCE product JASMES with a spatial resolution of 5 km throughout the Northern Hemisphere.

**9.Line 99, the sensor attenuation refers to AVHRR? Please specify it. Then, why the AVHRR sensor attenuation requires the algorithm to choose different TM images? And how different?**

**Response:** Thanks for your excellent question. Although AVHHR CDR products have been calibrated, the reflectance test of the Antarctic ice sheet at Dome-C (75°6' 0" S, 123°21'0" E) reveals that the attenuation is relatively low in the visible band but high in the near-infrared band, which leads to different product quality. The results are shown in figure (1), (2) and (3). Considering the difference in image data before and after 2000, we trained two sets of corresponding decision trees using different TM images, respectively.

[Figure]

Figure (1) The reflectance of band 1 from AVHRR SR V4 at Dome-C on the Antarctic ice sheet from 1981 to 2019.

[Figure]

Figure (2) The reflectance of band 2 from AVHRR SR V4 at Dome-C on the Antarctic ice sheet from 1981 to 2019.

[Figure]

Figure (3) The brightness temperature of BT11 from AVHRR SR V4 at Dome-C on the Antarctic ice sheet from 1981 to 2019.

**10. Line 103, where is the main seasonally snow-covered areas in China? Please specify it. This means move the content at Line 122 here.**

**Response:** Thanks for this insightful suggestion, and we have added the content of major seasonally

snow-covered regions in Line 102-104. We have also deleted the description of the three major snow regions in lines 120 to 121.

In order to ensure reliability and representativeness, the training and validating samples were evenly distributed in three major seasonally snow-covered regions across China, including North Xinjiang, Northeast China, and the Qinghai-Tibet Plateau.

The available CMA stations were evenly distributed across the three major seasonally snow-covered regions in China.

**11. Line 124, the usage of the snow depth data for the proposed study (line 127) should be used as the first sentence.**

**Response:** Thanks for your helpful suggestion, and we updated it as you suggested.

Che et al. (2008) and Dai et al. (2015) generated snow-depth data by using an inter-sensor calibration of multiple satellites' passive-microwave observations, which provides daily, 0.25-degree snow-depth data for China from 1979 to 2020. And this data set of long-term daily snow depth in China is available at http://data.tpdc.ac.cn.

**12. Line 128, the cloud/snow confusion is generated by averaging the hourly ERA 5 land climate reanalysis dataset?**

**Response:** Thanks for your helpful suggestion. Sorry! there is an obvious clerical error. The word "generated" in the sentence should be deleted.

We used the land surface temperature (LST) daily product to alleviate the cloud/snow confusion by averaging the hourly ERA 5 land climate reanalysis dataset on the GEE platform (Muñoz Sabater, 2019).

**13. Line 157, did you use the NDVI? If not, the formula (1) is not needed.**

**Response:** Thanks for your question. We do use NDVI in the cloud detection algorithm, as listed in Table 4. The cloud detection method considers elevation, NDVI, the difference of reflectance, and

brightness temperature. The cloud detection method could be grouped into A and B. A included 4 subcategories and B with 7 subcategories. Among of them, B7 adopts NDVI when identifying the non-snow category.

**14. Line 230, I suggest to add Huang as one of the co-authors.**

**Response:** Thanks for your advice. We just referenced a small part of the algorithm proposed by Huang (Huang et al., 2018). Huang was not involved in our work.

**15.Line 267-268, how to determine these thresholds?**

**Response:** A good question. Elevation classification refers to the MODIS global snow cover algorithm from NDIDC (Riggs et al., 2006). And the land surface temperature (LST) is obtained from our actual training samples. We count the distribution of BT11 (bright temperature in snow and no snow samples (Figure (4)), and use the BT11 condition (T4 <274.8 K) that can identify 95% of snow samples as the threshold, so we set one of the LST thresholds of the temperature mask in the paper to 275 K. The definition of 281 K for warm snow is derived from a large number of snow samples observed on the Tibetan Plateau.

[Figure]

Figure (4) The histogram of snow and snow-free pixels on BT 11 from AVHRR. The cyan samples are snow and the yellow samples are snow free.

Reference: Riggs, G.A., Hall, D.K., & Salomonson, V.V. (2006). MODIS Snow products user guide to collection 5. In. http://modis-snow-ice.gsfc.nasa.gov/?c=userguide

**16. Line 291, what did cause the poor quality of raw satellite?**

**Response:** A good question. The design of satellite sensor had limited presence life. There are inconsistencies in the calibration coefficients due to sensor changes at a later stage, which are the major reasons for the deterioration of data quality.

**17. Line 353, "poorly" is not good to describe the others' product.**

**Response:** Thanks for your helpful suggestion. The "poorly" has been revised to "much worse".

The JASMES SCE products performed much worse, with total OA, PA, UA, and CK values amounting to 71.8%, 39.2%, 68.1%, and 0.321, respectively.

**18. Line 375, what is the meaning of "GF"? Why not delete it?**

**Response:** Thanks for pointing this out. GF is the abbreviation of gap-filling. We have modified the abstract and body where gap-filled first appears.

Line 17-20: The NIEER AVHHR SCE product has the spatial resolution of 5-km and the daily temporal resolution, and is a completely gap-free product, which is produced through a series of processes such as the quality control, cloud detection, snow discrimination and gap-filling (GF).
Line 79-80: (5) Improved gap-filling (GF) strategies are adopted to obtain complete snow coverage.

**19.Line 379 and 380, why not list the names according to the increase of the values?**

**Response:** Thanks, it has been revised according to the order.

The values in the product are classified as non-snow (0), snow from AVHRR (1), snow from HMRF (2), snow from SD (3), water (4), and filling value (255).

**20. Line 383, as an ESSD paper, the conclusion should focus on the data, not the method.**

**Response:** Thanks for your helpful suggestion. The first paragraph of section "conclusions" has been revised according to the suggestion. We focused on the product and weakened the method.

Line428: In this study, a daily AVHRR SCE product with a spatial resolution of 5 km across China from 1981 to 2019 has been generated by the snow research team in the NIEER, Chinese Academy of Sciences. The NIEER AVHRR SCE product used a multi-level decision tree algorithm for cloud and snow discrimination and an improved GF technique. The product was validated using snow depth measurements provided by the China Meteorological Administration and higher spatial resolution SCE maps derived from Landsat-5 TM.

**21.For the other comments:**

**Line 22: "the producer's accuracy was 81.0% the user's accuracy was 81.3%, and the Cohen's kappa value was 0.717" --> "the producer's accuracy is 81.0%, the user's accuracy is 81.3%, and the Cohen's Kappa value is 0.717".**

**Line 47: "throughout China" should be deleted, because they are global products.**

**Line 52: "The polar orbit meteorological satellite of" should be deleted.**

**Line 84: "processing" -> "preprocessing".**

**Line 101: "true values" -> "ground truth".**

**Line 137: "between" should be deleted.**

**Line 139: "forth" -> "fourth".**

**Line 175: "previous" -> "the previous". It is the same in line 272.**

**Line 219: "algorithm above" -> "above-mentioned algorithm", and "first" should be deleted.**

**Line 270: "Methodology" -> "Metrics".**

**Line 315: delete "new".**

**Line 319: delete "the".**

**Line 321: "edges" -> "boundaries".**

**Line 327 and 328: "snow-covered areas" -> "snow cover".**

**Response:** All the above comments have been revised in the text of the new manuscript. Thank you again for all of the valuable comments or suggestions.

---

## Community Comment (CC2)

Response to Referee 2:

Thank you for your positive comments and interesting suggestions. This document intends to provide point-by-point answers to your remarks, directly proposing, where possible, modifications to the original paper that will be integrated into the revised version. We have worked in particular on:

• describing the discrimination algorithm of snow and cloud cover clearly.

• describing the discrimination algorithm of gap-filling strategies clearly.

• improving the language and writing styles

In the remainder of the document, lines in **bold** echo your comments for ease of reading, followed in the case by proposed modifications to our paper (with new elements in green).

We sincerely hope that these corrections will match your expectations.

**This paper employed a multi-level decision tree algorithm to detect cloud and snow based on AVHRR SR V4 data, then combined the HMRF-based spatio-temporal modeling technique and the snow-depth interpolation method to fill data gaps gradually. It produced a daily NIEER AVHRR SCE product with a spatial resolution of 5 km over China from 1981 to 2019. This product was validated using in situ snow depth measurements and SCE maps derived from Landsat-5 TM.**

**Since the cloud and snow confusion, as well as data gaps caused by cloud are common and occur often when mapping daily snow cover extent from optical sensors, techniques to solve these problems are valuable. The presented processing scheme was able to improve the quality of snow cover detection from AVHRR data. The produced long time-series snow cover extent product could be a significant dataset for studying climate change over China.**

**Despite of its significance, several issues still need to be resolved before a publication to ESSD. The quality control of AVHRR, why Landsat maps could be the true values to validate the cloud samples from AVHRR maps, and the three levels decision tree in flowchart could be sufficiently explained. In addition, it is not necessary to define a weight in HMRF modeling, since only one energy source was used in this study. Besides, the English of this paper should be further refined so as to improve the overall presentation.**

**1. P1, Abstract, change "15 percent" to "15%".**

**Response**: Thanks for your suggestion. In this article, we used two forms of percentage: one is 'number + %', like 60.8%, the other is 'number + percent', like 15 percent. The former represents the verification accuracy of the product; the latter represents the degree of accuracy improvement (relative degree) of the NIEER AVHRR SCE product compared to other products.

For example, compared with the well-known JASMES AVHRR product, the overall accuracy increased approximately 15 percent, the omission error dropped from nearly 60.8% to 19.7%, the commission error dropped from 31.9% to 21.3%, and the CK value increased by more than 114 percent.

**2. P2-3, it is suggested to provide a summary table of the mentioned SCE products, which lists the begin/end time, spatial resolution, temporal resolution, institution, produced methods, referenced paper, and download link.**

**Response:** Thanks for your suggestion. Currently, there are few snow cover products produced using AVHRR data, and the representative product is JASMES. At the same time, other snow products (NHSCE, IMS SCE, MODIS, and Fengyun SCE) mentioned in the article have significant differences in sensors, time\spatial resolution, and time\space coverage. And these products did not participate in the comparative experiment in Section 5.2. This article considers that only JASMES and NIEER AVHRR SCE are comparable, so only these two sets of data are used for accurate comparison. Therefore, this study does not provide a summary table of the mentioned SCE products.

**3. Figure 1, it only shows 7 Landsat senses for validation. Change "Landsat Snow Maps" to "Landsat senses for validation".**

**Response**: Thanks for your helpful suggestion. In the previous version of the article, we used a total of 8 verification images. Among them, there are two verification images in the Northern Xinjiang snow area that have the same spatial location and different image acquisition times, the row number is 147, the column number is 029, and the dates are respectively: 1997-02-17, 2016-02-22. So only 7 images are displayed in Figure 1.

However, to express the verification results more clearly, we have deleted and modified the used verification image in this version, excluding the image with the same spatial location in the Northern Xinjiang snow region, and added two verification images in the Northeast-Inner Mongolia snow region. We also added the major seasonal snow cover regions in Figure 1 (Using a white mask and a red border to indicate their range). More detailed information of all verified images is shown in the new Table 9.

First, the 9 images distributed in the three major snow-covered regions of China (3 images in Northern Xinjiang, 4 images in Northeast of China, and 2 images in the Qinghai-Tibet Plateau), which are very representative, as shown in Figure 1. Second, the main influencing factors for snow recognition are the type of land cover and topography. The snow-covered region of northern Xinjiang is mainly flat areas, in addition to the Tianshan Mountains and the Altai Mountains. We chose 3 sceneries, one scene is located in the flat area of Altay (C8), one scene is in the Altai Mountains (C7), and the other scene is in the Tianshan Mountains (C9). The northeast-Inner

Mongolia snow area is relatively flat, and the snow is mainly distributed in forest areas, cultivated land and grassland, so we choose two forest area images (C1, C3), located in the Greater Khingan Range area and Small Khingan Range area respectively, with one farmland (C2) and grassland in Inner Mongolia (C4). The snow cover of the Qinghai-Tibet Plateau is mainly distributed in mountainous areas. And the verification images we selected are mainly distributed in mountainous areas (C5, C6). The main types of land cover are grassland and bare land. Considering the above factors, the images we selected are sufficiently representative.

In Figure 1, "Landsat Snow Maps" has been changed to "Landsat senses for validation".

[Figure]

Figure 1: The geographic location of study area and the spatial distribution of major snow-covered regions, climate stations and Landsat-5 validation dataset. The elevation data were derived from Shuttle Radar Topography Mission (SRTM).

**Table 9 The accuracy of NIEER AVHRR SCE maps versus Landsat-5 TM SCE maps. C1~C9 denotes the different Landsat-5 TM SCE.**

| Path/row | Serial number | Date | Cloud percentage | Snow percentage | OA | PA | UA | CK |
|----------|---------------|------|------------------|-----------------|------|------|------|------|
| 116028 | C1 | 19970312 | 2.0% | 77.2% | 87.9% | 88.3% | 95.9% | 0.678 |
| 118029 | C2 | 20161109 | 0.2% | 88.0% | 84.5% | 82.8% | 99.5% | 0.519 |
| 121024 | C3 | 20160319 | 2.0% | 96.4% | 98.1% | 100.0% | 98.1% | 1 |
| 127031 | C4 | 20180130 | 1.1% | 45.3% | 82.0% | 63.0% | 96.1% | 0.626 |
| 135038 | C5 | 19961109 | 1.0% | 66.5% | 79.5% | 81.0% | 87.9% | 0.552 |

| | | | | | | | | |
|---|---|---|---|---|---|---|---|---|
| 137039 | C6 | 19961123 | 2.0% | 50.7% | 78.2% | 65.7% | 88.5% | 0.566 |
| 142027 | C7 | 19870323 | 0.0% | 96.1% | 97.2% | 100.0% | 97.2% | 0.036 |
| 143027 | C8 | 20051110 | 2.0% | 48.6% | 93.1% | 86.7% | 99.8% | 0.863 |
| 147029 | C9 | 20160222 | 1.1% | 89.0% | 90.6% | 91.4% | 98.0% | 0.587 |
| | | Total | | | 87.3% | 86.7% | 95.7% | 0.695 |

**4. P6, it is not clear that how to deal with these night/ dense dark vegetation /sunglint/ water/ cloud shadow/ cloudy/ unused pixels, according to the quality control information.**

**Response:** Thanks for your helpful suggestion. The description of how to control the quality of various pixels (Table 3) is not clear enough in section 3.1. We revised it to clearly indicate the bit flags used in this study and modify Table 3.

Only observations valid in all AVHRR channels were employed to directly generate SCE records by using the quality control bit flags of AVHRR SR V4. Table 3 shows all the quality control information from AVHRR SR V4 and the status of usage in this study. After quality control processing, the valid pixels were used as input for retrieval and the invalid pixels were regarded as gap pixels.

**Table 3: All the quality control information from AVHRR SR V4 and the status of usage in this study.**

| Bitmask | Description | Use or not |
|---|---|---|
| 15 | Polar flag (latitude over 60 degrees (land) or 50 degrees (ocean)) | No use |
| 14 | BRDF-correction issues | No use |
| 13 | RHO3 value is invalid | No use |
| 12 | Channel 5 value is invalid | Use |
| 11 | Channel 4 value is invalid | Use |
| 10 | Channel 3 value is invalid | Use |
| 9 | Channel 2 value is invalid | Use |
| 8 | Channel 1 value is invalid | Use |
| 7 | Channel 1-5 are valid | Use |
| 6 | Pixel is at night (height solar zenith) | Use |
| 5 | Pixel is over dense dark vegetation | No use |
| 4 | Pixel is over sunglint | No use |
| 3 | Pixel is over water | Use |
| 2 | Pixel contains cloud shadow | No use |
| 1 | Pixel is cloudy | No use |
| 0 | Unused | No use |

**5. P6, this paper used Landsat maps as the true values to validate the cloud and snow samples from AVHRR maps at the same days. It is okay for snow samples. However, for cloud samples,**

since cloud can change in a quite short time period, it depends on the overpass time of two satellites. Please provide more explanation.

Response: Thanks for this excellent question. For the matching problem of cloud samples between Landsat images and AVHRR images, we used the following two principles to avoid errors caused by cloud changes.

First, we select the training data used by the cloud recognition algorithm in non-snow period, which excludes the influence of snow on cloud recognition. Second, the selected training data is the Landsat image that is more than 80% covered by clouds. Due to the difference between AVHRR's transit time and Landsat's transit time, to ensure that the AVHRR image at the same spatial location on the corresponding date is also cloud-covered, we visually distinguish the selected samples to ensure that both are cloud-covered at the same time.

The following figure shows the comparison between the selected three scenes of Landsat cloud images and the corresponding AVHRR images. To ensure the accuracy of the AVHRR cloud recognition algorithm, all images selected have been visually interpreted as follows.

[Figure]

Figure (1): Cloud comparison of Landsat image with AVHRR images. (a) is located in Northeast China on Mar. 24st, 1992;(b) is located in Qinghai-Tibet Plateau on Feb. 28st, 2005; (c) is located in North Xinjiang on Jan. 09st, 1996.

**6. Figure 2, it is not clear for the three levels decision tree. For example, how about the hierarchical relationships among them? How about the input and output for each level?**

Response: Thanks for this insightful question. We have added a new snow discrimination flowchart (figure 4) to make the description of snow algorithm clearer. For each level of the decision tree, we have carried out a clear description. SR1, BT11 combined with DEM, and SR3/SR2, were chosen as first-level discriminators. The main purpose of the first-level decision tree is to exclude

pixels that are considered non-snow pixels. Figure (2) showed the frequency distribution of snow and non-snow of SR1 and BT11. According to the training results, the confidence level of snow samples is set as 95%, and the threshold value of the corresponding SR1 is 0.14. The pixels are classified as snow pixels when SR1>0.14, and the remaining pixels are classified as non-snow pixels. SR3/SR2, SR3-SR2, NDVI, and NDSI were compared, and SR3/SR2 was chosen as an auxiliary discriminant index for the first-level decision tree because of the lowest discrimination. In the same way, the threshold value of other indexes was obtained. After the first-level decision discrimination, the possible snow pixels are used as the input of the second-level decision tree.

[Figure]

**Figure (2) The frequency distribution histogram and optimal threshold acquisition of snow and snow free before 2000. (a) is the SR1 frequency distribution of snow and snow free on AVHRR, and (b) is the BT11 frequency distribution of snow and snow free on AVHRR**

The second-level decision tree is mainly used to obtain the pixels of determined snow among the potential snow pixels provided by the first-level decision tree. According to the ability to distinguish snow from the training samples, NDVI and SR3-SR2 are used as a discriminant index for the second-level decision tree. From Figure (3), the NDVI threshold is set to -0.16 at the confidence level of 99%. The pixel is identified as snow cover when NDVI <-0.16. The SR3-SR2 threshold is set to -0.81 at the confidence level of 99%. The pixel is identified as snow cover when SR3-SR2<-0.81. After the second-level decision discrimination, the pixels of uncertain type are used as the input of the third-level decision tree.

[Figure]

**Figure (3) Histograms of NDVI and SR3-SR2 snow and non-snow frequency distributions and discriminant thresholds,(a) NDVI;(b) SR3-SR2**

NDSI has the highest ability to detect snow cover pixels, which is considered as the third-level decision tree. Figure 5 in the paper described the method of optimal NDSI threshold. Same with optimal cloud test, the NDSI cross-point of snow and non-snow pixels frequency distribution were obtained by the highest overall accuracy of snow cover was calculated with the step of NDSI as

0.01. Finally, the pixels with NDSI > 0.73 are snow pixels, and those with NDSI ≤0.73 are non-snow pixels. The snow and non-snow pixels were finally merged to produce the snow data under clear sky conditions. Table 5 shows the snow discrimination scheme and thresholds before and after 2000.

The text in the paper is revised as follows:

To improve the snow discrimination under clear-skies, all decision rules were re-adjusted according to the training samples from high-resolution snow maps. We developed a three-level decision tree algorithm, which obtained the optimal threshold values from the training data. Using Landsat-5 TM data as true values, we obtained the frequency distribution characteristics of each band from AVHRR data in the snow and non-snow areas at SR1, BT11, SR3/SR2, SR3-SR2, NDVI, and NDSI. Figure 4 shows the flowchart of the three-level decision tree snow discrimination algorithm.

SR1, BT11 combined with DEM, and SR3/SR2, were chosen as first-level discriminators. The main purpose of the first-level decision tree is to exclude pixels that are definitely non-snow pixels. Snow has high reflectance in the SR1 band and low brightness temperature in the thermal infrared BT11 band. Since the ability to distinguish snow of SR3/SR2 is lower than SR3-SR2 by our training test, the SR3/SR2 was chosen as a first-level discriminator. Based on the frequency distributions of snow and non-snow pixels for the first-level discriminators for Landsat-5 TM maps, a confidence level of 99% of snow samples was set to obtain the threshold value of certain non-snow pixels. As shown in Table 5, for the samples before 2000, SR1 was >0.14 and BT11<274 K when DEM<1300 m, BT11 was <281 K when DEM≥1300 m, and SR3/SR2<0.50 were the possible snow images, while the remaining pixels were non-snow pixels. The potential snow pixels were used as input for the second-level decision tree.

NDVI and SR3-SR2 were chosen as second-level discriminators. The second-level decision tree was mainly used to obtain certain snow pixels from the possible snow pixels. Based on the frequency distributions of snow and non-snow pixels from potential snow pixels processed by the first-level decision tree, a confidence level of 99% of non-snow samples was set to obtain the threshold value of certain snow pixels. For the samples before 2000, a pixel was classified as certain snow when NDVI < -0.16 and SR3-SR2 < -0.81 (Table 5). Other pixels were considered the potential snow pixels, which were used as input for the third-level decision tree.

NDSI was used as the third-level discriminator due to its excellent discrimination ability of snow cover and other land covers. Based on the frequency distributions of potential snow pixels derived from the second-level decision tree, the optimal NDSI threshold value was calculated by a method similar to that of the cloud test. Figure 4 shows the optimal NDSI scheme. Fig.5 (a) presents the NDSI frequency distribution histogram of snow and non-snow pixels. The cross-point of snow and non-snow that has the highest overall accuracy (85.87%) was chosen as the optimal NDSI threshold (0.73), as shown in Fig 5(b). The cross-point also represents a compromise for the snow omission (15.83%) and commission error (13.03%). Thus, pixels with NDSI>0.73 were identified as snow for the samples before 2000.

[Figure]

**Figure 4: The flowchart of a three-level decision tree snow discrimination algorithm for NIEER AVHRR SCE product.**

[Figure]

**Figure 5: NDSI frequency distribution histogram and optimal threshold acquisition of snow and non-snow before 2000. (a) is the frequency distribution of snow and non-snow on AVHRR, and (b) is the optimal NDSI threshold value.**

**9. P9, the original HMRF snow framework integrates spectral information, spatio-temporal information, and environmental information to reclassify snow and non-snow classes. The total energy function includes each energy source and its optimal parameters to minimize the total energy function. Among them, the parameter indicate the contribution of corresponding energy source. The original HMRF modeling technique employs a cubic spatio-temporal neighborhood to represent the combination influence from temporal context and the spatial context, which is effective to fill the overwhelming majority of data gaps in MODIS snow cover products. This research only used the spatio-temporal information, it is not necessary to define**

**a weight for one energy source, as shown in equation 3. It is suggested to replace it by the spatio-temporal cubic energy function.**

**Response:** Thanks for your helpful suggestion. In this study, we only used the spatio-temporal information. The probability that a cloud may snow, snow-free, and undetermined under different spatio-temporal conditions was calculated. We have reworked the text and equations to convey this.

Here, we present a spatio-temporal modeling technique for filling up gap pixels in daily snow cover estimates based on the time series of AVHRR preliminary SCE records. The spatio-temporal modeling technique integrated AVHRR preliminary SCE record spatial and temporal contextual information within a Hidden Markov Random Field (HMRF) model (Melgani and Serpico, 2003). Initially, Huang et al. (2018) utilized HMRF based spectral information, spatio-temporal information, and environmental information to reclassify snow and non-snow classes by MODIS snow products. In our study, only used the spatio-temporal information for filling up gap pixels. The core of this method is computing the spatio-temporal cubic energy function for every gap from the neighborhood pixels and further classifying the gap pixels as snow pixels, non-snow pixels, or still gap pixels using

$$U_T(\beta_n) = U_{st}(\beta_n | N_{sp}, N_{tp}) \quad , \tag{3}$$

where $U_T$ is the total energy function of belonging to the class of $\beta_n$ (n=2, $\beta_1$ denotes snow and $\beta_2$ denotes non-snow), and $U_{st}$ is the spatio-temporal neighborhood cubic energy function. $N_{sp}$ and $N_{tp}$ denote the spatial neighborhood and temporal neighborhood centered with the gap pixel, respectively.

**10. P10-14, it is suggested to add some more thorough analysis and discussion. For example, the accuracy over North Xinjiang, Qinghai-Tibet Plateau, and Northeast China, the accuracy over different land cover types, as well as analyses and discussion with previous studies.**

**Response:** Thanks for your helpful suggestion. We add the major seasonally snow-covered regions in figure 7. From Figure 7, we can see the individual CMA validation results of the three major seasonally snow-covered regions in China. For the OA index, the accuracy of the Northern Xinjiang snow-covered region is better, followed by the Northeast of China snow-covered region. In terms of PA and UA indicators, the Northern Xinjiang snow-covered region has the best accuracy. The Northeast of China snow-covered area has high UA and low PA, indicating that the product underestimates the snow cover extent in this snow-covered region. And the accuracy of the Qinghai-Tibet plateau snow-covered region is relatively low, high PA and low UA indicate a misclassification phenomenon due to frequent instantaneous snow, and the range of snow cover varies significantly within a day. For the CK index, the accuracy of the Northern Xinjiang snow-covered area is better, followed by the Northeast of China snow-covered region, and the Qinghai-Tibet Plateau snow-covered region is poor. Therefore, it is unnecessary to discuss the accuracy of snow cover regions. The NIEER AVHRR SCE product has a relatively coarse spatial resolution of 5 km. The spatial coverage of one pixel involves several land cover types, making it hard to analyze the overall

accuracy quantitatively under a specific type. Therefore, this paper does not verify the accuracy of snow recognition under different land cover types.

[Figure]

**Figure 7: Point-based accuracy results of NIEER AVHRR product: (a) OA; (b) PA; (c) UA; (d) CK. The snow depth of 191 climate stations used is provided by the China Meteorological Administration (CMA). OA, PA, UA and CK represent overall accuracy, producer's accuracy, user's accuracy, and Cohen's Kappa coefficient.**

**12. Table 4, change "DEM<300" to "DEM≤300". Please provide more detailed information about the cloud detection and the corresponding thresholds. What are the clues to divide the Target A into A1-A4 and to divide Target B into B1-B10? All threshold were determined by tests?**

**Response:** Thanks for this excellent question. In this study, we adopted the cloud test scheme by Hori et al. (2017), but the critical threshold value of BT37-BT11 was adjusted. Cloud detection scheme described in section 3.2. The cloud test scheme is also derived from many previous studies (Hori et al., 2007; Stamnes et al., 2007; Yamanouchi et al., 1987). The cloud detection method could be grouped into A and B. A included 4 subcategories and B with 7 subcategories. As shown in table 4. The determination of the threshold of cloud is similar to that of snow. The cross-point of the snow and cloud frequency distribution curves represents the optimal threshold. Following the same procedure, the threshold of each type is calculated as shown in the figure below.

[Figure]

Figure2 Cloud and snow distribution histogram and optimal threshold acquisition of cloud detection.

**13. Table 5, please provide more information about the threshold values. How were they determined?**

**Response:** Thanks for this good question, the flowchart of threshold acquisition has been added to provide a detailed description of the three-level decision tree. Please refer to the answer to question 6 for details.

For the other comments:

7. Figure 3, it is suggested to change the text color of "snow" to green, and that of "cloud" to orange.

8. Figure 4, it is suggested to change the text color of "snow" to green, and that of "Non-Snow" to brown.

11. Table 2, change "Year" to "Time period".

14. Table 10, the snow column of JASMES SCE should be near the Non-snow column of JASMES SCE.

**Response:** All these remarks will be corrected in the revised text. Thanks very much again for your valuable implication and comments.

---

## Author Response (AR1)

Thank you for your positive comments and helpful suggestions. This document provides point-by-point answers to all reviews. We have revised the manuscript in particular on:

• describing the discrimination algorithm of snow and cloud clearly. A new flowchart process of snow discrimination algorithm was added.

• revised the discrimination algorithm of gap-filling strategies clearly. Revised and explain the equation.

• emphasizing the dataset and weakening the methodology in the Conclusion

• improving the language and writing styles

In the document, lines in **bold** echo your comments for ease of reading, the revised manuscript with new elements in green.

Responses to Referee 1:

This paper proposes an a long-term AVHRR snow cover extent product from 1981 until 2019 over China. The product has the spatial resolution of 5-km and the daily temporal resolution, and is a completely gap-free product, which is produced through quality control, cloud detection, snow discrimination, and gap-filling. The validations based on ground measurement and Landsat-5 snow maps both demonstrate its higher accuracy than that of the JASMES AVHRR product. As a long-term record, the dataset will provide a valuable data source for analyzing the influence of climate changes on the cryosphere on multiple time scales. The need for such a dataset is well justified and the authors cite ample relevant literature. The paper is basically well-written and presented.

1. it is best to delete "Using the Google Earth Engine (GEE) platform" in the first sentence because GEE is just the platform of producing the product, it is not the main contribution of the study. However, I suggest to add a subsection in the section of "data and preprocessing" to describe the computing platform and the reason of choosing GEE.

**Response: Thanks for this helpful suggestion. We have deleted "Using the Google Earth Engine (GEE) platform" in the first sentence, described the advantages of the GEE platform, and explained why the GEE platform is used.**

2. why not add the dataset of 2020 year?

**Response: Thanks for your suggestion. In the study, the NOAA Climate Data Record (CDR) of AVHRR Surface Reflectance Version 4 (AVHRR SR V4) was used as primary input data. The AVHHR CDR V4 used in our work is available from 1981-06-24T 00:00:00 to 2019-05-16T 00:00:00. Our work also indicated that the low quality of AVHHR after 2018 has a noticeable impact on the accuracy of snow cover extent products. This dataset developed mainly focused on the data before 2000. After 2019, the AVHRR CDR V4 product will no longer be produced due to satellite data quality issues. To extend the time series, we prefer to choose newly launched satellite sensors, like MODIS or NPP products.**

3. Is it sufficient enough to validate so big product using just eight Landsat-5 images?

**Response: We added the description of the three major snow-covered regions (Northern Xinjiang, Northeast China, and Qinghai-Tibet Plateau) of China in figure 1. In addition, we**

added two new validation images in Northeast China, and removed a similar validation image. A total of 9 Landsat-5 images distributed in the three major snow-covered regions were selected as validation data. And the detailed information of each verification image is shown in new table 9 (All serial number of images are rearranged). The main factors for snow cover detection are the type of land cover and topography. We chose three sceneries in Northern Xinjiang, one scene is located in the flat area of Altay (C8), one scene is in the Altai Mountains (C7), and the other scene is in the Tianshan Mountains (C9). The Northeast China, snow area is relatively flat. The snow is mainly distributed in forest areas, cultivated land and grassland, so we choose two forest area images (C1, C3), located in Greater Khingan Range area and Small Khingan Range area, respectively, with one farmland (C2) and grassland in Inner Mongolia (C4). The snow cover of the Qinghai-Tibet Plateau is mainly distributed in mountainous areas. And the verification data we selected are mainly distributed in mountainous areas (C5, C6). The main types of land cover are grassland and bare land. Considering the above factors, a total of 9 Landsat-5 validation images are representative and can meet the requirements.

[Figure]

Figure 1: The geographic location of study area and the spatial distribution of major snow-covered regions, climate stations and Lansat-5 validation dataset. The elevation data were derived from Shuttle Radar Topography Mission (SRTM).

Another validation with reference to higher-resolution snow maps derived from Landsat-5 Thematic Mapper (TM) images demonstrates an overall accuracy of 87.3%, a producer's accuracy of 86.7%, a user's accuracy of 95.7%, and a Cohen's kappa value of 0.695.

A total of 9 Landsat-5 TM snow maps were used as the validation dataset (Fig.1). The training and validating samples were evenly distributed across China's major seasonally snow-covered regions (including North Xinjiang, Northeast China, and the Qinghai-Tibet Plateau) to ensure reliability and representativeness.

In the study, 9 Landsat-5 snow maps were used to further evaluate the NIEER AVHRR product. Table 9 gives the validation results of our maps versus the Landsat-5 TM SCE maps. The OA was as high as 87.3%. The high UA and low PA revealed that the product has a slight tendency to underestimate the snow cover extent. The CK value (0.695) of the 'area to area' method also demonstrated 'substantial' agreement, which was close to that of ground measurements validation (0.717).

Figure 8 further displays three intuitional examples demonstrating the detailed difference between NIEER AVHRR SCE maps and Landsat-5 SCE reference maps. The three images (serial number "C1, C5, and C8") were located in Northeast China, the Qinghai-Tibet Plateau, and North Xinjiang, respectively.

Considering the limitations of point-to-area validation, the overall OA, PA, UA, and CK values were 87.3%, 86.7%, 95.7%, and 0.695, respectively, using Landsat-5 TM area-to-area, which showed the same trend of accuracy as the point validation.

**Table 9 The accuracy of NIEER AVHRR SCE maps versus Landsat-5 TM SCE maps. C1~C9 denotes the different Landsat-5 TM SCE.**

| Path/row | Serial number | Date | Cloud percentage | Snow percentage | OA | PA | UA | CK |
|---|---|---|---|---|---|---|---|---|
| 116028 | C1 | 19970312 | 2.0% | 77.2% | 87.9% | 88.3% | 95.9% | 0.678 |
| 118029 | C2 | 20161109 | 0.2% | 88.0% | 84.5% | 82.8% | 99.5% | 0.519 |
| 121024 | C3 | 20160319 | 2.0% | 96.4% | 98.1% | 100.0% | 98.1% | 1 |
| 127031 | C4 | 20180130 | 1.1% | 45.3% | 82.0% | 63.0% | 96.1% | 0.626 |
| 135038 | C5 | 19961109 | 1.0% | 66.5% | 79.5% | 81.0% | 87.9% | 0.552 |
| 137039 | C6 | 19961123 | 2.0% | 50.7% | 78.2% | 65.7% | 88.5% | 0.566 |
| 142027 | C7 | 19870323 | 0.0% | 96.1% | 97.2% | 100.0% | 97.2% | 0.036 |
| 143027 | C8 | 20051110 | 2.0% | 48.6% | 93.1% | 86.7% | 99.8% | 0.863 |
| 147029 | C9 | 20160222 | 1.1% | 89.0% | 90.6% | 91.4% | 98.0% | 0.587 |
| | | Total | | | 87.3% | 86.7% | 95.7% | 0.695 |

4.Line 17, NIEER is the abbreviation of the authors institute, "the new NIEER product" is difficult to project to the new AVHRR snow cover extent product, because many products are produced by the institute.

**Response: Thanks for your helpful suggestion. Referring to NASA NSIDC MODIS series, we plan to publish some snow products, including MODIS SCE, AVHRR SCE, and MODIS FSC. We revised "the new NIEER product" by "the NIEER AVHHR SCE products" to distinguish from the other products.**

5.Line 27, "nearly 40%" should be changed to precise number.

**Response: Thanks for the careful check. According to our validation results, we have changed**

**"nearly 40%" into "60.8%".**

6.Line 37, "these decades" refers to which decades? The citing references was published in 2005 and 2018.

**Response: Thanks for your helpful suggestion. We have replaced "over these decades" with "over the past several decades". Generally speaking, over these days, years or decades indicate during the past several days, years or decades.**

7.Line 38, "is not only of particular importance for climate research but also an indispensable indicator of climate change" needs to be rephrased.

**Response: Thanks for your suggestion. We improved this sentence by "Therefore, long-term snow cover data are particularly important for climate research and are also a crucial indicator of climate change."**

**8.Line 56, the abbreviation SCE has been described in the last paragraph.**

**Response: Thanks for the helpful suggestion, and we deleted it.**

9.Line 99, the sensor attenuation refers to AVHRR? Please specify it. Then, why the AVHRR sensor attenuation requires the algorithm to choose different TM images? And how different?

**Response: Thanks for your excellent question. Although AVHHR CDR products have been calibrated, the reflectance test of the Antarctic ice sheet at Dome-C (75°6' 0" S, 123°21'0" E) reveals that the attenuation is relatively low in the visible band but high in the near-infrared band, which leads to different product quality. The results are shown in figure (1), (2) and (3). Considering the difference in image data before and after 2000, we trained two sets of corresponding decision trees using different TM images, respectively.**

[Figure]

Figure (1) The reflectance of band 1 from AVHRR SR V4 at Dome-C on the Antarctic ice sheet from 1981 to 2019.

[Figure]

Figure (2) The reflectance of band 2 from AVHRR SR V4 at Dome-C on the Antarctic ice sheet from 1981 to 2019.

[Figure]

Figure (3) The brightness temperature of BT11 from AVHRR SR V4 at Dome-C on the Antarctic ice sheet from 1981 to 2019.

10. Line 103, where is the main seasonally snow-covered areas in China? Please specify it. This means move the content at Line 122 here.

**Response: Thanks for this insightful suggestion, and we have added the content of major seasonally snow-covered regions. We have also deleted the description of the three major snow regions.**

11. Line 124, the usage of the snow depth data for the proposed study (line 127) should be used as the first sentence.

**Response: Thanks for your helpful suggestion, and we updated it as you suggested.**

Che et al. (2008) and Dai et al. (2015) generated snow-depth data by using an inter-sensor calibration

of multiple satellites' passive-microwave observations, which provides daily, 0.25-degree snow-depth data for China from 1979 to 2020. And this data set of long-term daily snow depth in China is available at http://data.tpdc.ac.cn.

12. Line 128, the cloud/snow confusion is generated by averaging the hourly ERA 5 land climate reanalysis dataset?

**Response:Thanks for your helpful suggestion. Sorry! there is an obvious clerical error. The word "generated" in the sentence should be deleted.**

13. Line 157, did you use the NDVI? If not, the formula (1) is not needed.

**Response: Thanks for your question. We do use NDVI in the cloud detection algorithm, as listed in Table 4. The cloud detection method considers elevation, NDVI, the difference of reflectance, and brightness temperature. The cloud detection method could be grouped into A and B. A included 4 subcategories and B with 7 subcategories. Among of them, B7 adopts NDVI when identifying the non-snow category.**

14. Line 230, I suggest to add Huang as one of the co-authors.

**Response: Thanks for your advice. We just referenced a small part of the algorithm proposed by Huang (Huang et al., 2018). Huang was not involved in our work.**

15.Line 267-268, how to determine these thresholds?

**Response: A good question. Elevation classification refers to the MODIS global snow cover algorithm from NDIDC (Riggs et al., 2006). And the land surface temperature (LST) is obtained from our actual training samples. We count the distribution of BT11 (bright temperature in snow and no snow samples (Figure (4)), and use the BT11 condition (T4 <274.8 K) that can identify 95% of snow samples as the threshold, so we set one of the LST thresholds of the temperature mask in the paper to 275 K. The definition of 281 K for warm snow is derived from a large number of snow samples observed on the Tibetan Plateau.**

[Figure]

Figure (4) The histogram of snow and snow-free pixels on BT 11 from AVHRR. The cyan samples are snow and the yellow samples are snow free.

Reference: Riggs, G.A., Hall, D.K., & Salomonson, V.V. (2006). MODIS Snow products user guide to collection 5. In. http://modis-snow-ice.gsfc.nasa.gov/?c=userguide

16. Line 291, what did cause the poor quality of raw satellite?

**Response: A good question. The design of satellite sensor had limited presence life. There are inconsistencies in the calibration coefficients due to sensor changes at a later stage, which are the major reasons for the deterioration of data quality.**

17. Line 353, "poorly" is not good to describe the others' product.

**Response:Thanks for your helpful suggestion. The "poorly" has been revised to "much worse".**

The JASMES SCE products performed much worse, with total OA, PA, UA, and CK values amounting to 71.8%, 39.2%, 68.1%, and 0.321, respectively.

18. Line 375, what is the meaning of "GF"? Why not delete it?

**Response:Thanks for pointing this out. GF is the abbreviation of gap-filling. We have modified the abstract and body where gap-filled first appears.**

The NIEER AVHHR SCE product has the spatial resolution of 5-km and the daily temporal resolution, and is a completely gap-free product, which is produced through a series of processes such as the quality control, cloud detection, snow discrimination and gap-filling (GF).
Improved gap-filling (GF) strategies are adopted to obtain complete snow coverage.

19.Line 379 and 380, why not list the names according to the increase of the values?

**Response: Thanks, it has been revised according to the order.**

The values in the product are classified as non-snow (0), snow from AVHRR (1), snow from HMRF (2), snow from SD (3), water (4), and filling value (255).

**20. Line 383, as an ESSD paper, the conclusion should focus on the data, not the method.**

**Response:** Thanks for your helpful suggestion. The first paragraph of section "conclusions" has been revised according to the suggestion. We focused on the product and weakened the method.

Line428: In this study, a daily AVHRR SCE product with a spatial resolution of 5 km across China from 1981 to 2019 has been generated by the snow research team in the NIEER, Chinese Academy of Sciences. The NIEER AVHRR SCE product used a multi-level decision tree algorithm for cloud and snow discrimination and an improved GF technique. The product was validated using snow

depth measurements provided by the China Meteorological Administration and higher spatial resolution SCE maps derived from Landsat-5 TM.

21.For the other comments:

Line 22: "the producer's accuracy was 81.0% the user's accuracy was 81.3%, and the Cohen's kappa value was 0.717" --> "the producer's accuracy is 81.0%, the user's accuracy is 81.3%, and the Cohen's Kappa value is 0.717".

Line 47: "throughout China" should be deleted, because they are global products.

Line 52: "The polar orbit meteorological satellite of" should be deleted.

Line 84: "processing" -> "preprocessing".

Line 101: "true values" -> "ground truth".

Line 137: "between" should be deleted.

Line 139: "forth" -> "fourth".

Line 175: "previous" -> "the previous". It is the same in line 272.

Line 219: "algorithm above" -> "above-mentioned algorithm", and "first" should be deleted.

Line 270: "Methodology" -> "Metrics".

Line 315: delete "new".

Line 319: delete "the".

Line 321: "edges" -> "boundaries".

Line 327 and 328: "snow-covered areas" -> "snow cover".

**Response: All the above comments have been revised in the text of the new manuscript. Thank you again for all of the valuable comments or suggestions.**

Responses to Referee 2:

This paper employed a multi-level decision tree algorithm to detect cloud and snow based on AVHRR SR V4 data, then combined the HMRF-based spatio-temporal modeling technique and the snow-depth interpolation method to fill data gaps gradually. It produced a daily NIEER AVHRR SCE product with a spatial resolution of 5 km over China from 1981 to 2019. This product was validated using in situ snow depth measurements and SCE maps derived from Landsat-5 TM.

Since the cloud and snow confusion, as well as data gaps caused by cloud are common and occur often when mapping daily snow cover extent from optical sensors, techniques to solve these problems are valuable. The presented processing scheme was able to improve the quality of snow cover detection from AVHRR data. The produced long time-series snow cover extent product could be a significant dataset for studying climate change over China.

Despite of its significance, several issues still need to be resolved before a publication to ESSD. The quality control of AVHRR, why Landsat maps could be the true values to validate the cloud samples from AVHRR maps, and the three levels decision tree in flowchart could be sufficiently explained. In addition, it is not necessary to define a weight in HMRF modeling, since only one energy source was used in this study. Besides, the English of this paper should be further refined so as to improve the overall presentation.

1. P1, Abstract, change "15 percent" to "15%".
**Response: Thanks for your suggestion. In this article, we used two forms of percentage: one is 'number + %', like 60.8%, the other is 'number + percent', like 15 percent. The former represents the verification accuracy of the product; the latter represents the degree of accuracy improvement (relative degree) of the NIEER AVHRR SCE product compared to other products.**

2. P2-3, it is suggested to provide a summary table of the mentioned SCE products, which lists the begin/end time, spatial resolution, temporal resolution, institution, produced methods, referenced paper, and download link.
**Response: Thanks for your suggestion. Currently, there are few snow cover products produced using AVHRR data, and the representative product is JASMES. At the same time, other snow products (NHSCE, IMS SCE, MODIS, and Fengyun SCE) mentioned in the article have significant differences in sensors, time\spatial resolution, and time\space coverage. And these products did not participate in the comparative experiment in Section 5.2. This article considers that only JASMES and NIEER AVHRR SCE are comparable, so only these two sets of data are used for accurate comparison. Therefore, this study does not provide a summary table of the mentioned SCE products.**

3. Figure 1, it only shows 7 Landsat senses for validation. Change "Landsat Snow Maps" to "Landsat senses for validation".
**Response: Thanks for your helpful suggestion. In the previous version of the article, we used a total of 8 verification images. Among them, there are two verification images in the Northern**

**Xinjiang snow area that have the same spatial location and different image acquisition times, the row number is 147, the column number is 029, and the dates are respectively: 1997-02-17, 2016-02-22. So only 7 images are displayed in Figure 1. However, to express the verification results more clearly, we have deleted and modified the used verification image in this version, excluding the image with the same spatial location in the Northern Xinjiang snow region, and added two verification images in the Northeast-Inner Mongolia snow region. We also added the major seasonal snow cover regions in Figure 1 (Using a white mask and a red border to indicate their range). More detailed information of all verified images is shown in the new Table 9.**

4. P6, it is not clear that how to deal with these night/ dense dark vegetation /sunglint/ water/ cloud shadow/ cloudy/ unused pixels, according to the quality control information.

**Response: Thanks for your helpful suggestion. The description of how to control the quality of various pixels (Table 3) is not clear enough in section 3.1. We revised it to clearly indicate the bit flags used in this study and modify Table 3.**

5. P6, this paper used Landsat maps as the true values to validate the cloud and snow samples from AVHRR maps at the same days. It is okay for snow samples. However, for cloud samples, since cloud can change in a quite short time period, it depends on the overpass time of two satellites. Please provide more explanation.

**Response: Thanks for this excellent question. For the matching problem of cloud samples between Landsat images and AVHRR images, we used the following two principles to avoid errors caused by cloud changes.**

**First, we select the training data used by the cloud recognition algorithm in non-snow period, which excludes the influence of snow on cloud recognition. Second, the selected training data is the Landsat image that is more than 80% covered by clouds. Due to the difference between AVHRR's transit time and Landsat's transit time, to ensure that the AVHRR image at the same spatial location on the corresponding date is also cloud-covered, we visually distinguish the selected samples to ensure that both are cloud-covered at the same time.**

**The following figure shows the comparison between the selected three scenes of Landsat cloud images and the corresponding AVHRR images. To ensure the accuracy of the AVHRR cloud recognition algorithm, all images selected have been visually interpreted as follows.**

[Figure]

Figure (1): Cloud comparison of Landsat image with AVHRR images. (a) is located in Northeast China on Mar. 24st, 1992;(b) is located in Qinghai-Tibet Plateau on Feb. 28st, 2005; (c) is located in North Xinjiang on Jan. 09st, 1996.

**6. Figure 2, it is not clear for the three levels decision tree. For example, how about the hierarchical relationships among them? How about the input and output for each level?**

**Response:** Thanks for this insightful question. We have added a new snow discrimination flowchart (figure 4) to make the description of snow algorithm clearer. For each level of the decision tree, we have carried out a clear description. SR1, BT11 combined with DEM, and SR3/SR2, were chosen as first-level discriminators. The main purpose of the first-level decision tree is to exclude pixels that are considered non-snow pixels. Figure (2) showed the frequency distribution of snow and non-snow of SR1 and BT11. According to the training results, the confidence level of snow samples is set as 95%, and the threshold value of the corresponding SR1 is 0.14. The pixels are classified as snow pixels when SR1>0.14, and the remaining pixels are classified as non-snow pixels. SR3/SR2, SR3-SR2, NDVI, and NDSI were compared, and SR3/SR2 was chosen as an auxiliary discriminant index for the first-level decision tree because of the lowest discrimination. In the same way, the threshold value of other indexes was obtained. After the first-level decision discrimination, the possible snow pixels are used as the input of the second-level decision tree.

[Figure]

**Figure (2) The frequency distribution histogram and optimal threshold acquisition of snow and snow free before 2000. (a) is the SR1 frequency distribution of snow and snow free on AVHRR, and (b) is the BT11 frequency distribution of snow and snow free on AVHRR**

The second-level decision tree is mainly used to obtain the pixels of determined snow among the potential snow pixels provided by the first-level decision tree. According to the ability to distinguish snow from the training samples, NDVI and SR3-SR2 are used as a discriminant index for the second-level decision tree. From Figure (3), the NDVI threshold is set to -0.16 at the confidence level of 99%. The pixel is identified as snow cover when NDVI <-0.16. The SR3-SR2 threshold is set to -0.81 at the confidence level of 99%. The pixel is identified as snow cover when SR3-SR2<-0.81. After the second-level decision discrimination, the pixels of uncertain type are used as the input of the third-level decision tree.

[Figure]

**Figure (3) Histograms of NDVI and SR3-SR2 snow and non-snow frequency distributions and discriminant thresholds,(a) NDVI;(b) SR3-SR2**

NDSI has the highest ability to detect snow cover pixels, which is considered as the third-level decision tree. Figure 5 in the paper described the method of optimal NDSI threshold. Same with optimal cloud test, the NDSI cross-point of snow and non-snow pixels frequency distribution were obtained by the highest overall accuracy of snow cover was calculated with the step of NDSI as 0.01. Finally, the pixels with NDSI > 0.73 are snow pixels, and those with NDSI ≤0.73 are non-snow pixels. The snow and non-snow pixels were finally merged to produce the snow data under clear sky conditions. Table 5 shows the snow discrimination scheme and thresholds before and after 2000.

The text in the paper is revised as follows:

To improve the snow discrimination under clear-skies, all decision rules were re-adjusted according to the training samples from high-resolution snow maps. We developed a three-level decision tree algorithm, which obtained the optimal threshold values from the training data. Using Landsat-5 TM data as true values, we obtained the frequency distribution characteristics of each band from

AVHRR data in the snow and non-snow areas at SR1, BT11, SR3/SR2, SR3-SR2, NDVI, and NDSI. Figure 4 shows the flowchart of the three-level decision tree snow discrimination algorithm.

SR1, BT11 combined with DEM, and SR3/SR2, were chosen as first-level discriminators. The main purpose of the first-level decision tree is to exclude pixels that are definitely non-snow pixels. Snow has high reflectance in the SR1 band and low brightness temperature in the thermal infrared BT11 band. Since the ability to distinguish snow of SR3/SR2 is lower than SR3-SR2 by our training test, the SR3/SR2 was chosen as a first-level discriminator. Based on the frequency distributions of snow and non-snow pixels for the first-level discriminators for Landsat-5 TM maps, a confidence level of 99% of snow samples was set to obtain the threshold value of certain non-snow pixels. As shown in Table 5, for the samples before 2000, SR1 was >0.14 and BT11<274 K when DEM<1300 m, BT11 was <281 K when DEM≥1300 m, and SR3/SR2<0.50 were the possible snow images, while the remaining pixels were non-snow pixels. The potential snow pixels were used as input for the second-level decision tree.

NDVI and SR3-SR2 were chosen as second-level discriminators. The second-level decision tree was mainly used to obtain certain snow pixels from the possible snow pixels. Based on the frequency distributions of snow and non-snow pixels from potential snow pixels processed by the first-level decision tree, a confidence level of 99% of non-snow samples was set to obtain the threshold value of certain snow pixels. For the samples before 2000, a pixel was classified as certain snow when NDVI < -0.16 and SR3-SR2 < -0.81 (Table 5). Other pixels were considered the potential snow pixels, which were used as input for the third-level decision tree.

NDSI was used as the third-level discriminator due to its excellent discrimination ability of snow cover and other land covers. Based on the frequency distributions of potential snow pixels derived from the second-level decision tree, the optimal NDSI threshold value was calculated by a method similar to that of the cloud test. Figure 4 shows the optimal NDSI scheme. Fig.5 (a) presents the NDSI frequency distribution histogram of snow and non-snow pixels. The cross-point of snow and non-snow that has the highest overall accuracy (85.87%) was chosen as the optimal NDSI threshold (0.73), as shown in Fig 5(b). The cross-point also represents a compromise for the snow omission (15.83%) and commission error (13.03%). Thus, pixels with NDSI>0.73 were identified as snow for the samples before 2000.

[Figure]

**Figure 4: The flowchart of a three-level decision tree snow discrimination algorithm for NIEER AVHRR SCE product.**

[Figure]

**Figure 5: NDSI frequency distribution histogram and optimal threshold acquisition of snow and non-snow before 2000. (a) is the frequency distribution of snow and non-snow on AVHRR, and (b) is the optimal NDSI threshold value.**

9. P9, the original HMRF snow framework integrates spectral information, spatio-temporal information, and environmental information to reclassify snow and non-snow classes. The total energy function includes each energy source and its optimal parameters to minimize the total energy function. Among them, the parameter indicate the contribution of corresponding energy source. The original HMRF modeling technique employs a cubic spatio-temporal neighborhood to represent the combination influence from temporal context and the spatial context, which is effective to fill the overwhelming majority of data gaps in MODIS snow cover products. This research only used the spatio-temporal information, it is not necessary to define a weight for one energy source, as shown

in equation 3. It is suggested to replace it by the spatio-temporal cubic energy function.

**Response:Thanks for your helpful suggestion. In this study, we only used the spatio-temporal information. The probability that a cloud may snow, snow-free, and undetermined under different spatio-temporal conditions was calculated. We have reworked the text and equations to convey this.**

Here, we present a spatio-temporal modeling technique for filling up gap pixels in daily snow cover estimates based on the time series of AVHRR preliminary SCE records. The spatio-temporal modeling technique integrated AVHRR preliminary SCE record spatial and temporal contextual information within a Hidden Markov Random Field (HMRF) model (Melgani and Serpico, 2003). Initially, Huang et al. (2018) utilized HMRF based spectral information, spatio-temporal information, and environmental information to reclassify snow and non-snow classes by MODIS snow products. In our study, only used the spatio-temporal information for filling up gap pixels. The core of this method is computing the spatio-temporal cubic energy function for every gap from the neighborhood pixels and further classifying the gap pixels as snow pixels, non-snow pixels, or still gap pixels using

$$U_T(\beta_n) = U_{st}(\beta_n | N_{sp}, N_{tp})$$ , (3)

where $U_T$ is the total energy function of belonging to the class of $\beta_n$ (n=2, $\beta_1$ denotes snow and $\beta_2$ denotes non-snow), and $U_{st}$ is the spatio-temporal neighborhood cubic energy function. $N_{sp}$ and $N_{tp}$ denote the spatial neighborhood and temporal neighborhood centered with the gap pixel, respectively.

10. P10-14, it is suggested to add some more thorough analysis and discussion. For example, the accuracy over North Xinjiang, Qinghai-Tibet Plateau, and Northeast China, the accuracy over different land cover types, as well as analyses and discussion with previous studies.

**Response: Thanks for your helpful suggestion. We add the major seasonally snow-covered regions in figure 7. From Figure 7, we can see the individual CMA validation results of the three major seasonally snow-covered regions in China. For the OA index, the accuracy of the Northern Xinjiang snow-covered region is better, followed by the Northeast of China snow-covered region. In terms of PA and UA indicators, the Northern Xinjiang snow-covered region has the best accuracy. The Northeast of China snow-covered area has high UA and low PA, indicating that the product underestimates the snow cover extent in this snow-covered region. And the accuracy of the Qinghai-Tibet plateau snow-covered region is relatively low, high PA and low UA indicate a misclassification phenomenon due to frequent instantaneous snow, and the range of snow cover varies significantly within a day. For the CK index, the accuracy of the Northern Xinjiang snow-covered area is better, followed by the Northeast of China snow-covered region, and the Qinghai-Tibet Plateau snow-covered region is poor. Therefore, it is unnecessary to discuss the accuracy of snow cover regions.**

**The NIEER AVHRR SCE product has a relatively coarse spatial resolution of 5 km. The**

**spatial coverage of one pixel involves several land cover types, making it hard to analyze the overall accuracy quantitatively under a specific type. Therefore, this paper does not verify the accuracy of snow recognition under different land cover types.**

**12. Table 4, change "DEM<300" to "DEM≤300". Please provide more detailed information about the cloud detection and the corresponding thresholds. What are the clues to divide the Target A into A1-A4 and to divide Target B into B1-B10? All threshold were determined by tests?**

**Response:** Thanks for this excellent question. In this study, we adopted the cloud test scheme by Hori et al. (2017), but the critical threshold value of BT37-BT11 was adjusted. Cloud detection scheme described in section 3.2. The cloud test scheme is also derived from many previous studies (Hori et al., 2007; Stamnes et al., 2007; Yamanouchi et al., 1987). The cloud detection method could be grouped into A and B. An included 4 subcategories and B with 7 subcategories. As shown in table 4. The determination of the threshold of cloud is similar to that of snow. The cross-point of the snow and cloud frequency distribution curves represents the optimal threshold. Following the same procedure, the threshold of each type is calculated as shown in the figure below.

[Figure]

Figure2 **Cloud and snow distribution histogram and optimal threshold acquisition of cloud detection.**

13. Table 5, please provide more information about the threshold values. How were they determined?

**Response: Thanks for this good question, the flowchart of threshold acquisition has been added to provide a detailed description of the three-level decision tree. Please refer to the answer to question 6 for details.**

For the other comments:

7. Figure 3, it is suggested to change the text color of "snow" to green, and that of "cloud" to orange.

8. Figure 4, it is suggested to change the text color of "snow" to green, and that of "Non-Snow" to brown.

11. Table 2, change "Year" to "Time period".

14. Table 10, the snow column of JASMES SCE should be near the Non-snow column of JASMES SCE.

**Response: All these remarks will be corrected in the revised text. Thanks very much again for your valuable implication and comments.**

---

## Author Response (AR2)

**Responses to editor**

Dear editor,

We have revised our manuscript by your comments. We sincerely hope that these corrections can reach your expectations.

Point-by-point response:

Lines 111 to 113: typesetters will change text (thousand) to numbers here. Authors will need to ensure accuracy of conversions.

**Response: Thanks for your suggestion. We have provided the specific numbers and have ensured the accuracy of the numbers.**

The training samples before 2000 included 717,172 snow samples, 804,104 non-snow samples, and 82,904 cloud samples. Samples after 2000 included 7,304,310 snow samples, 8,394,959 non-snow samples, and 44,422 cloud samples.

Line 141: HRMF should be HMRF? Acronym need definition at first use.

**Response: Thanks for the careful check. HRMF have been modified to HMRF. It first appeared in line 141, so the full name was given. We also deleted the full name in line 233.**

Fourth, the gaps caused by clouds or invalid observations in the preliminary SCE record were filled with a set of gap-filling techniques, including Hidden Markov Random Field (HMRF)-based interpolation and snow-depth interpolation.

The spatio-temporal modeling technique integrated AVHRR preliminary SCE record spatial and temporal contextual information within a HMRF model (Melgani and Serpico, 2003).

Line 182: no need to capitalize 'Green'

**Response: The 'Green' has been revised by 'green'.**

The NDSI is usually calculated using the green (around a wavelength of 0.50μm) and shortwave infrared (around a wavelength of 1.60 μm) bands.

Please remove the S China Sea inset (9-dash area) in Figures 1, 8 and 12. Otherwise, Copernicus - as standard policy applied to all journals - will issue a disclaimer about these figures attached in prominent location to this manuscript. Particularly because this paper addresses snow cover, those tropical insets have no utility. Other investigators have successfully removed provocative map features.

**Response: Thanks very much again. All these figures have been corrected in the new manuscript.**

**Other modification:**

We also revised some wrong figure numbers (Fig.3, Fig.4 and Fig.5).